# A Non-Asymptotic Analysis of Oversmoothing in Graph Neural Networks

**Xinyi Wu**[1], **Zhengdao Chen**[2][*] **William Wang**[1], **Ali Jadbabaie**[1]
[1]Laboratory for Information and Decision Systems (LIDS), MIT
[2]Courant Institute of Mathematical Sciences, New York University
{xinyiwu,wwang314,jadbabai}@mit.edu, zc1216@nyu.edu

## Abstract

Oversmoothing is a central challenge of building more powerful Graph Neural Networks (GNNs). While previous works have only demonstrated that oversmoothing is inevitable when the number of graph convolutions tends to infinity, in this paper, we precisely characterize the mechanism behind the phenomenon via a non-asymptotic analysis. Specifically, we distinguish between two different effects when applying graph convolutions—an undesirable mixing effect that homogenizes node representations in different classes, and a desirable denoising effect that homogenizes node representations in the same class. By quantifying these two effects on random graphs sampled from the Contextual Stochastic Block Model (CSBM), we show that oversmoothing happens once the mixing effect starts to dominate the denoising effect, and the number of layers required for this transition is $O(\log N / \log(\log N))$ for sufficiently dense graphs with $N$ nodes. We also extend our analysis to study the effects of Personalized PageRank (PPR), or equivalently, the effects of initial residual connections on oversmoothing. Our results suggest that while PPR mitigates oversmoothing at deeper layers, PPR-based architectures still achieve their best performance at a shallow depth and are outperformed by the graph convolution approach on certain graphs. Finally, we support our theoretical results with numerical experiments, which further suggest that the oversmoothing phenomenon observed in practice can be magnified by the difficulty of optimizing deep GNN models.

## 1 Introduction

Graph Neural Networks (GNNs) are a powerful framework for learning with graph-structured data (Gori et al., 2005; Scarselli et al., 2009; Bruna et al., 2014; Duvenaud et al., 2015; Defferrard et al., 2016; Battaglia et al., 2016; Li et al., 2016). Most GNN models are built by stacking graph convolutions or *message-passing* layers (Gilmer et al., 2017), where the representation of each node is computed by recursively aggregating and transforming the representations of its neighboring nodes. The most representative and popular example is the Graph Convolutional Network (GCN) (Kipf & Welling, 2017), which has demonstrated success in *node classification*, a primary graph task which asks for node labels and identifies community structures in real graphs.

Despite these achievements, the choice of depth for these GNN models remains an intriguing question. GNNs often achieve optimal classification performance when networks are shallow. Many widely used GNNs such as the GCN are no deeper than 4 layers (Kipf & Welling, 2017; Wu et al., 2019), and it has been observed that for deeper GNNs, repeated message-passing makes node representations in different classes indistinguishable and leads to lower node classification accuracy—a phenomenon known as *oversmoothing* (Kipf & Welling, 2017; Li et al., 2018; Klicpera et al., 2019; Wu et al., 2019; Oono & Suzuki, 2020; Chen et al., 2020a;b; Keriven, 2022). Through the insight that graph convolutions can be regarded as low-pass filters on graph signals, prior studies have established that oversmoothing is inevitable when the number of layers in a GNN increases to infinity (Li et al., 2018; Oono & Suzuki, 2020). However, these asymptotic analyses do not fully explain the rapid occurrence of oversmoothing when we increase the network depth, let alone the fact that for some datasets, having no graph convolution is even optimal (Liu et al., 2021). These observations motivate the following key questions about oversmoothing in GNNs:

---

[*]Now at Google.

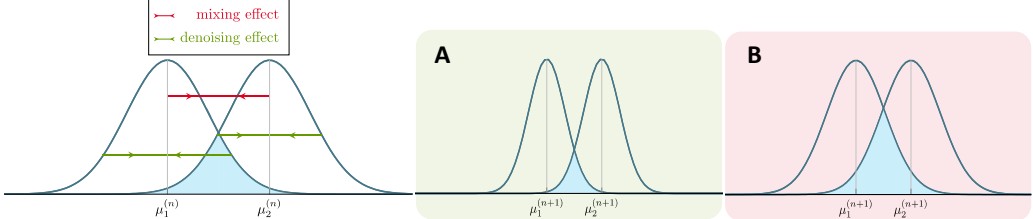

Figure 1: Stacking GNN layers increases both the mixing and denoising effects counteracting each other. Depending on the graph properties, either the denoising effect dominates the mixing effect, resulting in less difficulty classifying nodes (**A**), or the mixing effect dominates the denoising effect, resulting in more difficulty classifying nodes (**B**)—this is when oversmoothing starts to happen.

*Why does oversmoothing happen at a relatively shallow depth?*

*Can we quantitatively model the effect of applying a finite number of graph convolutions and theoretically predict the "sweet spot" for the choice of depth?*

In this paper, we propose a non-asymptotic analysis framework to study the effects of graph convolutions and oversmoothing using the *Contextual Stochastic Block Model (CSBM)* (Deshpande et al., 2018). The CSBM mimics the community structure of real graphs and enables us to evaluate the performance of linear GNNs through the probabilistic model with ground truth community labels. More importantly, as a generative model, the CSBM gives us full control over the graph structure and allows us to analyze the effect of graph convolutions non-asymptotically. In particular, we distinguish between two counteracting effects of graph convolutions:

- **mixing effect (undesirable)**: homogenizing node representations in different classes;
- **denoising effect (desirable)**: homogenizing node representations in the same class.

Adding graph convolutions will increase both the mixing and denoising effects. As a result, oversmoothing happens not just because the mixing effect keeps accumulating as the depth increases, on which the asymptotic analyses are based (Li et al., 2018; Oono & Suzuki, 2020), but rather because the mixing effect starts to dominate the denoising effect (see Figure 1 for a schematic illustration). By quantifying both effects as a function of the model depth, we show that the turning point of the tradeoff between the two effects is $O(\log N / \log(\log N))$ for graphs with $N$ nodes sampled from the CSBM in sufficiently dense regimes. Besides new theory, this paper also presents numerical results directly comparing theoretical predictions and empirical results. This comparison leads to new insights highlighting the fact that the oversmoothing phenomenon observed in practice is often a mixture of pure oversmoothing and difficulty of optimizing weights in deep GNN models.

In addition, we apply our framework to analyze the effects of Personalized PageRank (PPR) on oversmoothing. Personalized propagation of neural predictions (PPNP) and its approximate variant (APPNP) make use of PPR and its approximate variant, respectively, and were proposed as a solution to mitigate oversmoothing while retaining the ability to aggregate information from larger neighborhoods in the graph (Klicpera et al., 2019). We show mathematically that PPR makes the model performance more robust to increasing number of layers by reducing the mixing effect at each layer, while it nonetheless reduces the desirable denoising effect at the same time. For graphs with a large size or strong community structure, the reduction of the denoising effect would be greater than the reduction of the mixing effect and thus PPNP and APPNP would perform worse than the vanilla GNN on those graphs.

**Our contributions are summarized as follows:**

- We show that adding graph convolutions strengthens the denoising effect while exacerbates the mixing effect. Oversmoothing happens because the mixing effect dominates the denoising effect beyond a certain depth. For sufficiently dense CSBM graphs with $N$ nodes, the required number of layers for this to happen is $O(\log N / \log(\log N))$.

- We apply our framework to rigorously characterize the effects of PPR on oversmoothing. We show that PPR reduces both the mixing effect and the denoising effect of message-passing and thus does not necessarily improve node classification performance.

- We verify our theoretical results in experiments. Through comparison between theory and experiments, we find that the difficulty of optimizing weights in deep GNN architectures often aggravates oversmoothing.

## 2 ADDITIONAL RELATED WORK

**Oversmoothing problem in GNNs** Oversmoothing is a well-known issue in deep GNNs, and many techniques have been proposed to relieve it practically (Xu et al., 2018; Li et al., 2019; Chen et al., 2020b; Huang et al., 2020; Zhao & Akoglu, 2020). On the theory side, prior works have shown that as the model depth goes to infinity, the node representations within each connected component of the graph will converge to the same values (Li et al., 2018; Oono & Suzuki, 2020). However, The early onset of oversmoothing renders it an important concern in practice, and it has not been satisfyingly explained by the previous asymptotic studies. Our work addresses this gap by quantifying the effects of graph convolutions as a function of model depth and justifying why oversmoothing happens in shallow GNNs. A recent study shared a similar insight of distinguishing between two competing effects of message-passing and showed the existence of an optimal number of layers for node prediction tasks on a latent space random graph model. But the result had no further quantification on the optimal depth and hence the oversmoothing phenomenon was still only characterized asymptotically (Keriven, 2022).

**Analysis of GNNs on CSBMs** Stochastic block models (SBMs) and their contextual counterparts have been widely used to study node classification problems (Abbe, 2018; Chen et al., 2019). Recently there have been several works proposing to use CSBMs to theoretically analyze GNNs for the node classification task. Wei et al. (2022) used CSBMs to study the function of nonlinearity on the node classification performance, while Fountoulakis et al. (2022) used CSBMs to study the attention-based GNNs. More relevantly, Baranwal et al. (2021; 2022) showed the advantage of applying graph convolutions up to three times for node classification on CSBM graphs. Nonetheless, they only focused on the desirable denoising effect of graph convolution instead of its tradeoff with the undesirable mixing effect, and therefore did not explain the occurrence of oversmoothing.

## 3 PROBLEM SETTING AND MAIN RESULTS

We first introduce our theoretical analysis setup using the Contextual Stochastic Block Model (CSBM), a random graph model with planted community structure (Deshpande et al., 2018; Baranwal et al., 2021; 2022; Ma et al., 2022; Wei et al., 2022; Fountoulakis et al., 2022). We then present a set of theoretical results establishing bounds for the representation power of GNNs in terms of the best-case node classification accuracy. The proofs of all the theorems and additional claims will be provided in the Appendix.

### 3.1 NOTATIONS

We represent an undirected graph with $N$ nodes by $\mathcal{G} = (A, X)$, where $A \in \{0, 1\}^{N \times N}$ is the *adjacency matrix* and $X \in \mathbb{R}^N$ is the *node feature vector*. For nodes $u, v \in [N]$, $A_{uv} = 1$ if and only if $u$ and $v$ are connected with an edge in $\mathcal{G}$, and $X_u \in \mathbb{R}$ represents the node feature of $u$. We let $\mathbb{1}_N$ denote the all-one vector of length $N$ and $D = \text{diag}(A\mathbb{1}_N)$ be the *degree matrix* of $\mathcal{G}$.

### 3.2 THEORETICAL ANALYSIS FRAMEWORK

**Contextual Stochastic Block Models** We will focus on the case where the CSBM consists of two classes $\mathcal{C}_1$ and $\mathcal{C}_2$ of nodes of equal size, in total with $N$ nodes. For any two nodes in the graph, if they are from the same class, they are connected by an edge independently with probability $p$, or if they are from different classes, the probability is $q$. For each node $v \in \mathcal{C}_i, i \in \{1, 2\}$, the initial feature $X_v$ is sampled independently from a Gaussian distribution $\mathcal{N}(\mu_i, \sigma^2)$, where $\mu_i \in \mathbb{R}, \sigma \in (0, \infty)$. Without loss of generality, we assume that $\mu_1 < \mu_2$. We denote a graph generated from such a CSBM as $\mathcal{G}(A, X) \sim \text{CSBM}(N, p, q, \mu_1, \mu_2, \sigma^2)$. We further impose the following assumption on the CSBM used in our analysis.

**Assumption 1.** $p, q = \omega(\log N/N)$ *and* $p > q > 0$.

The choice $p, q = \omega(\log N/N)$ ensures that the generated graph $\mathcal{G}$ is connected almost surely (Abbe, 2018) while being slightly more general than the $p, q = \omega(\log^2 N/N)$ regime considered in some concurrent works (Baranwal et al., 2021; Wei et al., 2022). In addition, this regime also guarantees that $\mathcal{G}$ has a small diameter. Real-world graphs are known to exhibit the "small-world" phenomenon—even if the number of nodes $N$ is very large, the diameter of graph remains small (Girvan & Newman, 2002; Chung, 2010). We will see in the theoretical analysis (Section 3.3) how this small-diameter characteristic contributes to the occurrence of oversmoothing in shallow

GNNs. We remark that our results in fact hold for the more general choice of $p, q = \Omega(\log N/N)$, for which only the concentration bound in Theorem 1 needs to be modified in the threshold $\log N/N$ case where all the constants need a more careful treatment.

Further, the choice $p > q$ ensures that the graph structure has *homophily*, meaning that nodes from the same class are more likely to be connected than nodes from different classes. This characteristic is observed in a wide range of real-world graphs (Easley & Kleinberg, 2010; Ma et al., 2022). We note that this homophily assumption ($p > q$) is not essential to our analysis, though we add it for simplicity since the discussion of homophily versus heterophily ($p < q$) is not the focus of our paper.

**Graph convolution and linear GNN**  In this paper, our theoretical analysis focuses on the simplified linear GNN model defined as follows: a *graph convolution* using the (left-)normalized adjacency matrix takes the operation $h' = (D^{-1}A)h$, where $h$ and $h'$ are the input and output node representations, respectively. A linear GNN layer can then be defined as $h' = (D^{-1}A)hW$, where $W$ is a learnable weight matrix. As a result, the output of $n$ linear GNN layers can be written as $h^{(n)} \prod_{k=1}^{n} W^{(k)}$, where $h^{(n)} = (D^{-1}A)^n X$ is the output of $n$ graph convolutions, and $W^{(k)}$ is the weight matrix of the $k^{\text{th}}$ layer. Since this is linear in $h^{(n)}$, it follows that $n$-layer linear GNNs have the equivalent representation power as linear classifiers applied to $h^{(n)}$.

In practice, when building GNN models, nonlinear activation functions can be added between consecutive linear GNN layers. For additional results showing that adding certain nonlinearity would not improve the classification performance, see Appendix K.1.

**Bayes error rate and z-score**  Thanks to the linearity of the model, we see that the representation of node $v \in \mathcal{C}_i$ after $n$ graph convolutions is distributed as $\mathcal{N}(\mu_i^{(n)}, (\sigma^{(n)})^2)$, where the variance $(\sigma^{(n)})^2$ is shared between classes. The optimal node-wise classifier in this case is the *Bayes optimal classifier*, given by the following lemma.

**Lemma 1.** *Suppose the label $y$ is drawn uniformly from $\{1, 2\}$, and given $y$, $x \sim \mathcal{N}(\mu_y^{(n)}, (\sigma^{(n)})^2)$. Then the Bayes optimal classifier, which minimizes the probability of misclassification among all classifiers, has decision boundary $\mathcal{D} = (\mu_1 + \mu_2)/2$, and predicts $y = 1$, if $x \leq \mathcal{D}$ or $y = 2$, if $x > \mathcal{D}$. The associated Bayes error rate is $1 - \Phi(z^{(n)})$, where $\Phi$ denotes the cumulative distribution function of the standard Gaussian distribution and $z^{(n)} = \frac{1}{2}(\mu_2^{(n)} - \mu_1^{(n)})/\sigma^{(n)}$ is the z-score of $\mathcal{D}$ with respect to $\mathcal{N}(\mu_1^{(n)}, (\sigma^{(n)})^2)$.*

Lemma 1 states that we can estimate the optimal performance of an $n$-layer linear GNN through the z-score $z^{(n)} = \frac{1}{2}(\mu_2^{(n)} - \mu_1^{(n)})/\sigma^{(n)}$. A *higher* z-score indicates a *smaller* Bayes error rate, and hence a better expected performance of node classification. The z-score serves as a basis for our quantitative analysis of oversmoothing. In the following section, by estimating $\mu_2^{(n)} - \mu_1^{(n)}$ and $(\sigma^{(n)})^2$, we quantify the two counteracting effects of graph convolutions and obtain bounds on the z-score $z^{(n)}$ as a function of $n$, which allows us to characterize oversmoothing quantitatively. Specifically, there are two potential interpretations of oversmoothing based on the z-score: (1) $z^{(n)} < z^{(n^\star)}$, where $n^\star = \arg\max_{n'} z^{(n')}$; and (2) $z^{(n)} < z^{(0)}$. They correspond to the cases (1) $n > n^\star$; and (2) $n > n_0$, where $n_0 \geq 0$ denotes the number of layers that yield a z-score on par with $z^{(0)}$. The bounds on the z-score $z^{(n)}$, $z_{\text{lower}}^{(n)}$ and $z_{\text{upper}}^{(n)}$, enable us to estimate $n^\star$ and $n_0$ under different scenarios and provide insights into the optimal choice of depth.

## 3.3 Main Results

We first estimate the gap between the means $\mu_2^{(n)} - \mu_1^{(n)}$ with respect to the number of layers $n$. $\mu_2^{(n)} - \mu_1^{(n)}$ measures how much node representations in different classes have homogenized after $n$ GNN layers, which is the undesirable mixing effect.

**Lemma 2.** *For $n \in \mathbb{N} \cup \{0\}$, assuming $D^{-1}A \approx \mathbb{E}[D]^{-1}\mathbb{E}[A]$,*

$$\mu_2^{(n)} - \mu_1^{(n)} = \left(\frac{p - q}{p + q}\right)^n (\mu_2 - \mu_1).$$

Lemma 2 states that the means $\mu_1^{(n)}$ and $\mu_2^{(n)}$ get closer exponentially fast and as $n \to \infty$, both $\mu_1^{(n)}$ and $\mu_2^{(n)}$ will converge to the same value (in this case $(\mu_1^{(n)} + \mu_2^{(n)})/2$). The rate of change

$(p - q)/(p + q)$ is determined by the intra-community edge density $p$ and the inter-community edge density $q$. Lemma 2 suggests that graphs with higher inter-community density ($q$) or lower intra-community density ($p$) are expected to suffer from a higher mixing effect when we perform message-passing. We provide the following concentration bound for our estimate of $\mu_2^{(n)} - \mu_1^{(n)}$, which states that the estimate concentrates at a rate of $O(1/\sqrt{N(p + q)})$.

**Theorem 1.** *Fix $K \in \mathbb{N}$ and $r > 0$. There exists a constant $C(r, K)$ such that with probability at least $1 - O(1/N^r)$, it holds for all $1 \leq k \leq K$ that*

$$|(\mu_2^{(k)} - \mu_1^{(k)}) - \left(\frac{p - q}{p + q}\right)^k (\mu_2 - \mu_1)| \leq \frac{C}{\sqrt{N(p + q)}}.$$

We then study the variance $(\sigma^{(n)})^2$ with respect to the number of layers $n$. The variance $(\sigma^{(n)})^2$ measures how much the node representations in the same class have homogenized, which is the desirable denoising effect. We first state that no matter how many layers are applied, there is a nontrivial fixed lower bound for $(\sigma^{(n)})^2$ for a graph with $N$ nodes.

**Lemma 3.** *For all $n \in \mathbb{N} \cup \{0\}$, $\frac{1}{N}\sigma^2 \leq (\sigma^{(n)})^2 \leq \sigma^2$ .*

Lemma 3 implies that for a given graph, even as the number of layers $n$ goes to infinity, the variance $(\sigma^{(n)})^2$ does not converge to zero, meaning that there is a fixed lower bound for the denoising effect. See Appendix K.2 for the exact theoretical limit for the variance $(\sigma^{(n)})^2$ as $n$ goes to infinity. We now establish a set of more precise upper and lower bounds for the variance $(\sigma^{(n)})^2$ with respect to the number of layers $n$ in the following technical lemma.

**Lemma 4.** *Let $a = Np/\log N$. With probability at least $1 - O(1/N)$, it holds for all $1 \leq n \leq N$ that*

$$\max\left\{\frac{\min\{a, 2\}}{10} \frac{1}{(Np)^n}, \frac{1}{N}\right\}\sigma^2 \leq (\sigma^{(n)})^2$$

$$(\sigma^{(n)})^2 \leq \min\left\{\sum_{k=0}^{\lfloor\frac{n}{2}\rfloor} \frac{9}{\min\{a, 2\}}(n - 2k + 1)^{2k}(Np)^{n-2k}\left(\frac{2}{N(p + q)}\right)^{2n-2k}, 1\right\}\sigma^2 .$$

Lemma 4 holds for all $1 \leq n \leq N$ and directly leads to the following theorem with a clarified upper bound where $n$ is bounded by a constant $K$.

**Theorem 2.** *Let $a = Np/\log N$. Fix $K \in \mathbb{N}$. There exists a constant $C(K)$ such that with probability at least $1 - O(1/N)$, it holds for all $1 \leq n \leq K$ that*

$$\max\left\{\frac{\min\{a, 2\}}{10} \frac{1}{(Np)^n}, \frac{1}{N}\right\}\sigma^2 \leq (\sigma^{(n)})^2 \leq \min\left\{\frac{C}{\min\{a, 2\}} \frac{1}{(N(p + q))^n}, 1\right\}\sigma^2 .$$

Theorem 2 states that the variance $(\sigma^{(n)})^2$ for each Gaussian distribution decreases more for larger graphs or denser graphs. Moreover, the upper bound implies that the variance $(\sigma^{(n)})^2$ will initially go down at least at a rate exponential in $O(1/\log N)$ before reaching the fixed lower bound $\sigma^2/N$ suggested by Lemma 3. This means that after $O(\log N/\log(\log N))$ layers, the desirable denoising effect homogenizing node representations in the same class will saturate and the undesirable mixing effect will start to dominate.

**Why does oversmoothing happen at a shallow depth?** For each node, message-passing with different-class nodes homogenizes their representations exponentially. The exponential rate depends on the fraction of different-class neighbors among all neighbors (Lemma 2, mixing effect). Meanwhile, message-passing with nodes that have not been encountered before causes the denoising effect, and the magnitude depends on the absolute number of newly encountered neighbors. The diameter of the graph is approximately $\log N/\log(Np)$ in the $p, q = \Omega(\log N/N)$ regime (Graham & Lu, 2001), and thus is at most $\log N/\log(\log N)$ in our case. After the number of layers surpasses the diameter, for each node, there will be no nodes that have not been encountered before in message-passing and hence the denoising effect will almost vanish (Theorem 2, denoising effect). $\log N/\log(\log N)$ grows very slowly with $N$; for example, when $N = 10^6$, $\log N/\log(\log N) \approx 8$. This is why even in a large graph, the mixing effect will quickly dominate the denoising effect when we increase the number of layers, and so oversmoothing is expected to happen at a shallow depth.

Our theory suggests that the optimal number of layers, $n^\star$, is at most $O(\log N/\log(\log N))$. For a more quantitative estimate, we can use Lemma 2 and Lemma 4 to compute bounds $z_{\text{lower}}^{(n)}$ and

$z_{\text{upper}}^{(n)}$ for $z = \frac{1}{2}(\mu_2^{(n)} - \mu_1^{(n)})/\sigma^{(n)}$ and use them to infer $n^\star$ and $n_0$, as defined in Section 3.2. See Appendix H for detailed discussion.

Next, we investigate the effect of increasing the dimension of the node features $X$. So far, we have only considered the case with one-dimensional node features. The following proposition states that if features in each dimension are independent, increasing input feature dimension decreases the Bayes error rate for a fixed $n$. The intuition is that when node features provide more evidence for classification, it is easier to classify nodes correctly.

**Proposition 1.** *Let the input feature dimension be $d$, $X \in \mathbb{R}^{N \times d}$. Without loss of generality, suppose for node $v$ in $\mathcal{C}_i$, initial node feature $X_v \sim \mathcal{N}([\mu_i]^d, \sigma^2 I_d)$ independently. Then the Bayes error rate is $1 - \Phi\left(\frac{\sqrt{d}}{2}\frac{(\mu_2^{(n)} - \mu_1^{(n)})}{\sigma^{(n)}}\right) = 1 - \Phi\left(\frac{\sqrt{d}}{2}z^{(n)}\right)$, where $\Phi$ denotes the cumulative distribution function of the standard Gaussian distribution. Hence the Bayes error rate is decreasing in $d$, and as $d \to \infty$, it converges to $0$.*

## 4 THE EFFECTS OF PERSONALIZED PAGERANK ON OVERSMOOTHING

Our analysis framework in Section 3.3 can also be applied to GNNs with other message-passing schemes. Specifically, we can analyze the performance of Personalized Propagation of Neural Predictions (PPNP) and its approximate variant, Approximate PPNP (APPNP), which were proposed for alleviating oversmoothing while still making use of multi-hop information in the graph. The main idea is to use Personalized PageRank (PPR) or the approximate Personalized PageRank (APPR) in place of graph convolutions (Klicpera et al., 2019). Mathematically, the output of PPNP can be written as $h^{\text{PPNP}} = \alpha(I_N - (1 - \alpha)(D^{-1}A))^{-1}X$, while APPNP computes $h^{\text{APPNP}(n+1)} = (1 - \alpha)(D^{-1}A)h^{\text{APPNP}(n)} + \alpha X$ iteratively in $n$, where $I_N$ is the identity matrix of size $N$ and in both cases $\alpha$ is the teleportation probability. Then for nodes in $\mathcal{C}_i, i \in \{1, 2\}$, the node representations follow a Gaussian distribution $\mathcal{N}\left(\mu_i^{\text{PPNP}}, (\sigma^{\text{PPNP}})^2\right)$ after applying PPNP, or a Gaussian distribution $\mathcal{N}\left(\mu_i^{\text{APPNP}(n)}, (\sigma^{\text{APPNP}(n)})^2\right)$ after applying $n$ APPNP layers.

We quantify the effects on the means and variances for PPNP and APPNP in the CSBM case. We can similarly use them to calculate the z-score of $(\mu_1 + \mu_2)/2$ and compare it to the one derived for the baseline GNN in Section 3. The key idea is that the PPR propagation can be written as a weighted average of the standard message-passing, i.e. $\alpha(I_N - (1 - \alpha)(D^{-1}A))^{-1} = \sum_{k=0}^{\infty}(1 - \alpha)^k(D^{-1}A)^k$ (Andersen et al., 2006). We first state the resulting mixing effect measured by the difference between the two means.

**Proposition 2.** *Fix $r > 0, K \in \mathbb{N}$. For PPNP, with probability at least $1 - O(1/N^r)$, there exists a constant $C(\alpha, r, K)$ such that*

$$\mu_2^{PPNP} - \mu_1^{PPNP} = \frac{p+q}{p + \frac{2-\alpha}{\alpha}q}(\mu_2 - \mu_1) + \epsilon.$$

*where the error term $|\epsilon| \leq C/\sqrt{N(p+q)} + (1-\alpha)^{K+1}$.*

**Proposition 3.** *Let $r > 0$. For APPNP, with probability at least $1 - O(1/N^r)$,*

$$\mu_2^{APPNP(n)} - \mu_1^{APPNP(n)} = \left(\frac{p+q}{p + \frac{2-\alpha}{\alpha}q} + \frac{(2-2\alpha)q}{\alpha p + (2-\alpha)q}(1-\alpha)^n\left(\frac{p-q}{p+q}\right)^n\right)(\mu_2 - \mu_1) + \epsilon.$$

*where the error term $\epsilon$ is the same as the one defined in Theorem 1 for the case of $K = n$.*

Both $\frac{p+q}{p + \frac{2-\alpha}{\alpha}q}$ and $\frac{(2-2\alpha)q}{\alpha p + (2-\alpha)q}(1-\alpha)\left(\frac{p-q}{p+q}\right)$ are monotone increasing in $\alpha$. Hence from Proposition 2 and 3, we see that with larger $\alpha$, meaning a higher probability of teleportation back to the root node at each step of message-passing, PPNP and APPNP will indeed make the difference between the means of the two classes larger: while the difference in means for the baseline GNN decays as $\left(\frac{p-q}{p+q}\right)^n$, the difference for PPNP/APPNP is lower bounded by a constant. This validates the original intuition behind PPNP and APPNP that compared to the baseline GNN, they reduce the mixing effect of message-passing, as staying closer to the root node means aggregating less information from nodes of different classes. This advantage becomes more prominent when $n$ is larger, where the model performance is dominated by the mixing effect: as $n$ tends to infinity, while the means converge to the same value for the baseline GNN, their separation is lower-bounded for PPNP/APPNP.

However, the problem with the previous intuition is that PPNP and APPNP will also reduce the denoising effect at each layer, as staying closer to the root node also means aggregating less information from new nodes that have not been encountered before. Hence, for an arbitrary graph, the result of the tradeoff after the reduction of both effects is not trivial to analyze. Here, we quantify the resulting denoising effect for CSBM graphs measured by the variances. We denote $(\sigma^{(n)})^2_{\text{upper}}$ as the variance upper bound for depth $n$ in Lemma 4.

**Proposition 4.** *For PPNP, with probability at least $1 - O(1/N)$, it holds for all $1 \leq K \leq N$ that*

$$\max\left\{\frac{\alpha^2 \min\{a, 2\}}{10}, \frac{1}{N}\right\} \sigma^2 \leq (\sigma^{PPNP})^2 \leq \max\left\{\alpha^2 \left(\sum_{k=0}^{K} (1-\alpha)^k \sqrt{(\sigma^{(k)})^2_{upper}} + \frac{(1-\alpha)^{K+1}}{\alpha} \sigma\right)^2, \sigma^2\right\}.$$

**Proposition 5.** *For APPNP, with probability at least $1 - O(1/N)$, it holds for all $1 \leq n \leq N$ that*

$$\max\left\{\frac{\min\{a, 2\}}{10}\left(\alpha^2 + \frac{(1-\alpha)^{2n}}{(Np)^n}\right), \frac{1}{N}\right\} \sigma^2 \leq (\sigma^{APPNP(n)})^2,$$

$$(\sigma^{APPNP(n)})^2 \leq \min\left\{\left(\alpha\left(\sum_{k=0}^{n-1} (1-\alpha)^k \sqrt{(\sigma^{(k)})^2_{upper}}\right) + (1-\alpha)^n \sqrt{(\sigma^{(n)})^2_{upper}}\right)^2, \sigma^2\right\}.$$

By comparing the lower bounds in Proposition 4 and 5 with that in Theorem 2, we see that PPR reduces the beneficial denoising effect of message-passing: for large or dense graphs, while the variances for the baseline GNN decay as $1/(Np)^n$, the variances for PPNP/APPNP are lower bounded by the constant $\alpha^2 \min\{a, 2\}/10$. In total, the mixing effect is reduced by a factor of $\left(\frac{p-q}{p+q}\right)^n$, while the denoising effect is reduced by a factor of $1/(Np)^n$. Hence PPR would cause greater reduction in the denoising effect than the improvement in the mixing effect for graphs where $N$ and $p$ are large. This drawback would be especially notable at a shallow depth, where the denoising effect is supposed to dominate the mixing effect. APPNP would perform worse than the baseline GNN on these graphs in terms of the optimal classification performance.

We remark that in each APPNP layer, another way to interpret the term $\alpha X$ is to regard it as a residual connection to the initial representation $X$ (Chen et al., 2020b). Thus, our theory also validates the empirical observation that adding initial residual connections allows us to build very deep models without catastrophic oversmoothing. However, our results suggest that initial residual connections do not guarantee an improvement in model performance by themselves.

## 5 EXPERIMENTS

In this section, we first demonstrate our theoretical results in previous sections on synthetic CSBM data. Then we discuss the role of optimizing weights $W^{(k)}$ in GNN layers in the occurrence of oversmoothing through both synthetic data and the three widely used benchmarks: Cora, CiteSeer and PubMed (Yang et al., 2016). Our results highlight the fact that the oversmoothing phenomenon observed in practice can be exacerbated by the difficulty of optimizing weights in deep GNN models. More details about the experiments are provided in Appendix J.

### 5.1 THE EFFECT OF GRAPH TOPOLOGY ON OVERSMOOTHING

We first show how graph topology affects the occurrence of oversmoothing and the effects of PPR. We randomly generated synthetic graph data from CSBM($N = 2000$, $p$, $q = 0.0038$, $\mu_1 = 1$, $\mu_2 = 1.5, \sigma^2 = 1$). We used $60\%/20\%/20\%$ random splits and ran GNN and APPNP with $\alpha = 0.1$. For results in Figure 2, we report averages over 5 graphs and for results in Figure 3, we report averages over 5 runs.

In Figure 2, we study how the strength of community structure affects oversmoothing. We can see that when graphs have a stronger community structure in terms of a higher intra-community edge density $p$, they would benefit more from repeated message-passing. As a result, given the same set of node features, oversmoothing would happen later and a classifier could achieve better classification performance. A similar trend can also be observed in Figure 4A. Our theory predicts $n^\star$ and $n_0$, as defined in Section 3.2, with high accuracy.

In Figure 3, we compare APPNP and GNN under different graph topologies. In all three cases, APPNP manifests its advantage of reducing the mixing effect compared to GNN when the number

of layers is large, i.e. when the undesirable mixing effect is dominant. However, as Figure 3B,C show, when we have large graphs or graphs with strong community structure, APPNP's disadvantage of concurrently reducing the denoising effect is more severe, particularly when the number of layers is small. As a result, APPNP's optimal performance is worse than the baseline GNN. These observations accord well with our theoretical discussions in Section 4.

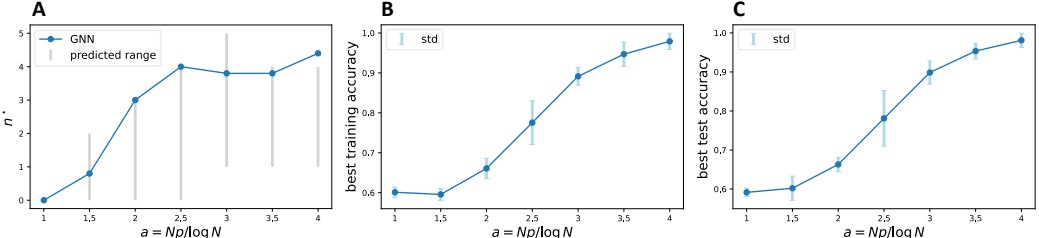

Figure 2: How the strength of community structure affects oversmoothing. When graphs have stronger community structure (i.e. higher $a$), oversmoothing would happen later. Our theory (gray bar) predicts the optimal number of layers $n^\star$ in practice (blue) with high accuracy (**A**). Given the same set of features, a classifier has significantly better performance on graphs with higher $a$ (**B,C**).

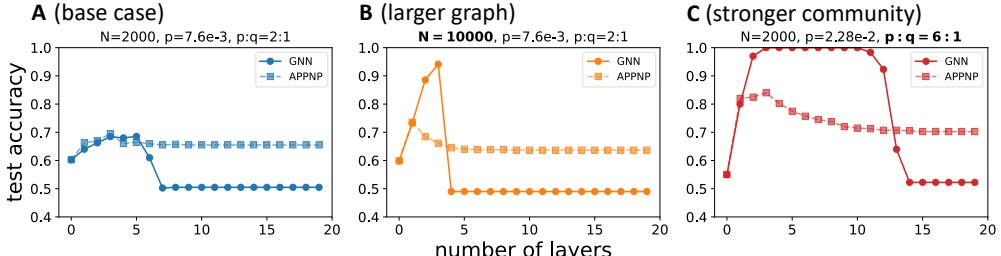

Figure 3: Comparison of node classification performance between the baseline GNN and APPNP. The performance of APPNP is more robust when we increase the model depth. However, compared to the base case (**A**), APPNP tends to have worse optimal performance than GNN on graphs with larger size (**B**) or stronger community structure (**C**), as predicted by the theory.

## 5.2 THE EFFECT OF OPTIMIZING WEIGHTS ON OVERSMOOTHING

We investigate how adding learnable weights $W^{(k)}$ in each GNN layers affects the node classification performance in practice. Consider the case when all the GNN layers used have width one, meaning that the learnable weight matrix $W^{(k)}$ in each layer is a scalar. In theory, the effects of adding such weights on the means and the variances would cancel each other and therefore they would not affect the z-score of our interest and the classification performance. Figure 4A shows the the value of $n_0$ predicted by the z-score, the actual $n_0$ without learnable weights according to the test accuracy and the actual $n_0$ with learnable weights according to the test accuracy. The results are averages over 5 graphs for each case. We empirically observe that GNNs with weights are much harder to train, and the difficulty increases as we increase the number of layers. As a result, $n_0$ is smaller for the model with weights and the gap is larger when $n_0$ is supposed to be larger, possibly due to greater difficulty in optimizing deeper architectures (Shamir, 2019).

To relieve this potential optimization problem, we increase the width of each GNN layer (Du & Hu, 2019). Figure 4B,C presents the training and testing accuracies of GNNs with increasing width with respect to the number of layers on a specific synthetic example. The results are averages over 5 runs. We observe that increasing the width of the network mitigates the difficulty of optimizing weights, and the performance after adding weights is able to gradually match the performance without weights. This empirically validates our claim in Section 3.2 that adding learnable weights should not affect the representation power of GNN in terms of node classification accuracy on CSBM graphs, besides empirical optimization issues.

In practice, as we build deeper GNNs for more complicated tasks on real graph data, the difficulty of optimizing weights in deep GNN models persists. We revisit the multi-class node classification task on the three widely used benchmark datasets: Cora, CiteSeer and PubMed (Yang et al., 2016). We compare the performance of GNN without weights against the performance of GNN with weights

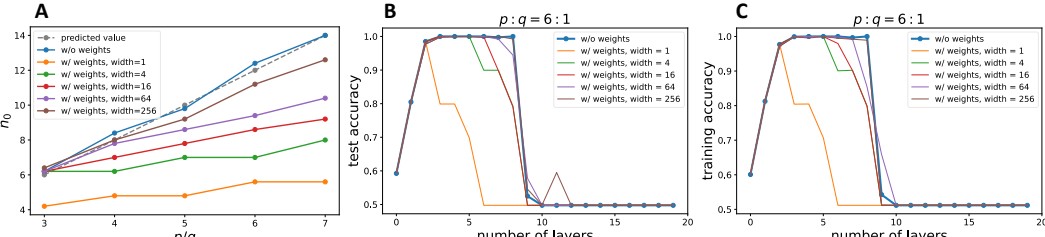

Figure 4: The effect of optimizing weights on oversmoothing using synthetic CSBM data. Compared to the GNN without weights, oversmoothing happens much sooner after adding learnable weights in each GNN layer, although these two models have the same representation power (**A**). As we increase the width of each GNN layer, the performance of GNN with weights is able to gradually match that of GNN without weights (**B,C**).

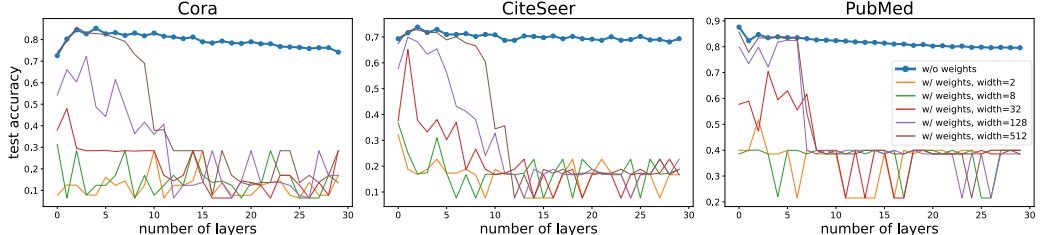

Figure 5: The effect of optimizing weights on oversmoothing using real-world benchmark datasets. Adding learnable weights in each GNN layer does not improve node classification performance but rather leads to optimization difficulty.

in terms of test accuracy. We used $60\%/20\%/20\%$ random splits, as in Wang & Leskovec (2020) and Huang et al. (2021) and report averages over 5 runs. Figure 5 shows the same kind of difficulty in optimizing deeper models with learnable weights in each GNN layer as we have seen for the synthetic data. Increasing the width of each GNN layer still mitigates the problem for shallower models, but it becomes much more difficult to tackle beyond 10 layers to the point that simply increasing the width could not solve it. As a result, although GNNs with and without weights are on par with each other when both are shallow, the former has much worse performance when the number of layers goes beyond 10. These results suggest that the oversmoothing phenomenon observed in practice is aggravated by the difficulty of optimizing deep GNN models.

## 6 DISCUSSION

Designing more powerful GNNs requires deeper understanding of current GNNs—how they work and why they fail. In this paper, we precisely characterize the mechanism of overmoothing via a non-asymptotic analysis and justify why oversmoothing happens at a shallow depth. Our analysis suggests that oversmoothing happens once the undesirable mixing effect homogenizing node representations in different classes starts to dominate the desirable denoising effect homogenizing node representations in the same class. Due to the small diameter characteristic of real graphs, the turning point of the tradeoff will occur after only a few rounds of message-passing, resulting in oversmoothing in shallow GNNs.

It is worth noting that oversmoothing becomes an important problem in the literature partly because typical Convolutional Neural Networks (CNNs) used for image processing are much deeper than GNNs (He et al., 2016). As such, researchers have been trying to use methods that have previously worked for CNNs to make current GNNs deeper (Li et al., 2019; Chen et al., 2020b). However, images can be regarded as giant grids with high diameter. This contrasts with with real-world graphs, which often have much smaller diameters. Hence we believe that building more powerful GNNs will require us to think beyond CNNs and images and take advantage of the structure in real graphs.

There are many outlooks to our work and possible directions for further research. First, while our use of the CSBM provided important insights into GNNs, it will be helpful to incorporate other real graph properties such as degree heterogeneity in the analysis. Additionally, further research can focus on the learning perspective of the problem.

ACKNOWLEDGMENTS

This research has been supported by a Vannevar Bush Fellowship from the Office of the Secretary of Defense. The U.S. Government is authorized to reproduce and distribute reprints for Governmental purposes notwithstanding any copyright annotation thereon. The views and conclusions contained herein are those of the authors and should not be interpreted as necessarily representing the official policies or endorsements, either expressed or implied, of the Department of Defense or the U.S. Government.

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

## A  PROOF OF LEMMA 1

Following the definition of the Bayes optimal classifier (Devroye et al., 1996),

$$\mathcal{B}(x) = \arg\max_{i=1,2} \quad \mathbb{P}[y = i | x],$$

we get that the Bayes optimal classifier has a linear decision boundary $\mathcal{D} = (\mu_1 + \mu_2)/2$ such that the decision rule is

$$\begin{cases} y = 1 & \text{if } x \leq \mathcal{D} \\ y = 2 & \text{if } x > \mathcal{D}. \end{cases}$$

Probability of misclassification could be written as

$$\mathbb{P}[y = 1, x > \mathcal{D}] + \mathbb{P}[y = 2, x \leq \mathcal{D}] = \mathbb{P}[x > \mathcal{D} | y = 1]\mathbb{P}[y = 1] + \mathbb{P}[x \leq \mathcal{D} | y = 2]\mathbb{P}[y = 2]$$

$$= \frac{1}{2}\left(\mathbb{P}[x > \mathcal{D} | y = 1] + \mathbb{P}[x \leq \mathcal{D} | y = 2]\right).$$

When $\mathcal{D} = (\mu_1 + \mu_2)/2$, the expression is called the Bayes error rate, which is the minimal probability of misclassification among all classifiers. Geometrically, it is easy to see that the Bayes error rate equals $\frac{1}{2}S$, where $S$ is the overlapping area between the two Gaussian distributions $\mathcal{N}\left(\mu_1^{(n)}, (\sigma^{(n)})^2\right)$ and $\mathcal{N}\left(\mu_2^{(n)}, (\sigma^{(n)})^2\right)$. Hence one can use the z-score of $(\mu_1 + \mu_2)/2$ with respect to either of the two Gaussian distributions to directly calculate the Bayes error rate.

## B  PROOF OF LEMMA 2

Under the heuristic assumption $D^{-1}A \approx \mathbb{E}[D]^{-1}\mathbb{E}[A]$, we can write

$$\mu_1^{(1)} = \frac{p\mu_1 + q\mu_2}{p + q}, \quad \mu_2^{(1)} = \frac{p\mu_2 + q\mu_1}{p + q}$$

$$\mu_1^{(k)} = \frac{p\mu_1^{(k-1)} + q\mu_2^{(k-1)}}{p+q}, \quad \mu_2^{(k)} = \frac{p\mu_2^{(k-1)} + q\mu_1^{(k-1)}}{p+q}, \text{ for all } k \in \mathbb{N}.$$

Writing recursively, we get that

$$\mu_1^{(n)} = \frac{(p+q)^n + (p-q)^n}{2(p+q)^n}\mu_1 + \frac{(p+q)^n - (p-q)^n}{2(p+q)^n}\mu_2,$$

$$\mu_2^{(n)} = \frac{(p+q)^n + (p-q)^n}{2(p+q)^n}\mu_2 + \frac{(p+q)^n - (p-q)^n}{2(p+q)^n}\mu_1.$$

## C  PROOF OF THEOREM 1

We use $\|\cdot\|_2$ to denote the spectral norm, $\|A\|_2 = \max_{x:\|x\|=1}\|Ax\|_2$. We denote $\bar{A} = \mathbb{E}[A]$, $\bar{D} = \mathbb{E}[D]$, $d = A\mathbb{1}_N$ and $\bar{d} = \mathbb{E}[d]_i$. We further define the following relevant vectors:

$$w_1 := \mathbb{1}_N, \quad w_2 := \begin{pmatrix} \mathbb{1}_{N/2} \\ -\mathbb{1}_{N/2} \end{pmatrix}, \quad \mu := \begin{pmatrix} \mu_1 \mathbb{1}_{N/2} \\ \mu_2 \mathbb{1}_{N/2} \end{pmatrix}.$$

The quantity of interest is $\mu_2^{(k)} - \mu_1^{(k)} = \frac{1}{N/2} w_2^\top (D^{-1}A)^k \mu$.

### C.1  AUXILIARY RESULTS

We record some properties of the adjacency matrices:

1. $D^{-1}A$ and $\bar{D}^{-1}\bar{A}$ have an eigenvalue of 1, corresponding to the (right) eigenvector $w_1$.
2. If $J_n = \mathbb{1}_n \mathbb{1}_n^\top$, where $\mathbb{1}_n$ is all-one vector of length $n$, then

$$\bar{A} := \begin{pmatrix} pJ_{N/2} & qJ_{N/2} \\ qJ_{N/2} & pJ_{N/2} \end{pmatrix}.$$

3. $\bar{D} = \frac{N}{2}(p+q)I_N$.
4. $\mu = \alpha w_1 + \beta w_2$, where $\alpha = \frac{\mu_1 + \mu_2}{2}$ and $\beta = \frac{\mu_1 - \mu_2}{2}$.

To control the degree matrix $D^{-1}$, we will use the following standard Chernoff bound Chung & Lu (2006):

**Lemma 5** (Chernoff Bound). *Let $X_1, ..., X_n$ be independent, $S := \sum_{i=1}^n X_i$, and $\bar{S} = \mathbb{E}[S]$. Then for all $\varepsilon > 0$,*

$$\mathbb{P}(S \leq \bar{S} - \varepsilon) \leq e^{-\varepsilon^2/(2\bar{S})},$$

$$\mathbb{P}(S \geq \bar{S} + \varepsilon) \leq e^{-\varepsilon^2/(2(\bar{S}+\varepsilon/3))}.$$

We can thus derive a uniform lower bound on the degree of every vertex:

**Corollary 1.** *For every $r > 0$, there is a constant $C(r)$ such that whenever $\bar{d} \geq C \log N$, with probability at least $1 - N^{-r}$,*

$$\frac{1}{2}\bar{d} \leq d_i \leq \frac{3}{2}\bar{d}, \quad \text{for all } 1 \leq i \leq N.$$

*Consequently, with probability at least $1 - N^{-r}$, $\|D^{-1} - \bar{D}^{-1}\|_2 \leq C/\bar{d}$ for some $C$.*

*Proof.* By applying Lemma 1 and a union bound, all degrees are within $1/2\bar{d}$ of their expectations, with probability at least $1 - e^{-\bar{d}/8 + \log N}$. Taking $C = 8r + 8$ yields the desired lower bound. An analogous proof works for the upper bound.

To show the latter part, write

$$\|D^{-1} - \bar{D}^{-1}\|_2 = \max_{1 \leq i \leq N} \frac{|d_i - \bar{d}|}{d_i \bar{d}}$$

Using the above bounds, the numerator for each $i$ is at most $1/2\bar{d}$ and the denominator for each $i$ is at least $1/2\bar{d}^2$, with probability at least $1 - N^{-r}$. Combining the bounds yields the claim. $\square$

We will also need a result on concentration of random adjacency matrices, which is a corollary of the sharp bounds derived in Bandeira & Van Handel (2016)

**Lemma 6** (Concentration of Adjacency Matrix). *For every $r > 0$, there is a constant $C(r)$ such that whenever $\bar{d} \geq \log N$, with probability at least $1 - N^{-r}$,*

$$\|A - \bar{A}\|_2 < C\sqrt{\bar{d}}.$$

*Proof.* By corollary 3.12 from Bandeira & Van Handel (2016), there is a constant $\kappa$ such that

$$\mathbb{P}(\|A - \bar{A}\|_2 \geq 3\sqrt{\bar{d}} + t) \leq e^{-t^2/\kappa + \log N}.$$

Setting $t = \sqrt{(1+r)\bar{d}}$, $C = 3 + \sqrt{(1+r)\kappa}$ suffices to achieve the desired bound. □

## C.2 SHARP CONCENTRATION OF THE RANDOM WALK OPERATOR $D^{-1}A$

In this section, we aim to show the following concentration result for the random walk operator $D^{-1}A$:

**Theorem 3.** *Suppose the edge probabilities are $\omega\left(\frac{\log N}{N}\right)$, and let $\bar{d}$ be the average degree. For any $r$, there exists a constant $C$ such that for sufficiently large $N$, with probability at least $1 - O(N^{-r})$,*

$$\|D^{-1}A - \bar{D}^{-1}\bar{A}\|_2 \leq \frac{C}{\sqrt{\bar{d}}}.$$

*Proof.* We decompose the error

$$E = D^{-1}A - \bar{D}^{-1}\bar{A} = D^{-1}(A - \bar{A}) + (D^{-1} - \bar{D}^{-1})\bar{A} = T_1 + T_2,$$

where

$$T_1 = D^{-1}(A - \bar{A}), \quad T_2 = (D^{-1} - \bar{D}^{-1})\bar{A}.$$

We bound the two terms separately.

**Bounding $T_1$:** By Corollary 1, $\|D^{-1}\|_2 = \max_i 1/d_i \leq 2/\bar{d}$ with probability $1 - N^{-r}$. Combining this with Lemma 6, we see that with probability at least $1 - 2N^{-r}$,

$$\|D^{-1}(A - \bar{A})\|_2 \leq \|D^{-1}\|_2 \|A - \bar{A}\|_2 \leq \frac{C}{\sqrt{\bar{d}}}$$

for some $C$ depending only on $r$.

**Bounding $T_2$:** Similar to Lu & Peng (2013), we bound $T_2$ by exploiting the low-rank structure of the expected adjacency matrix, $\bar{A}$. Recall that $\bar{A}$ has a special block form. The eigendecomposition of $\bar{A}$ is thus

$$\bar{A} = \sum_{j=1}^{2} \lambda_j w^{(j)},$$

where $w^{(1)} = \frac{1}{\sqrt{N}}\mathbb{1}_N, \lambda_1 = \frac{N(p+q)}{2}, w^{(2)} = \frac{1}{\sqrt{N}}\begin{pmatrix} \mathbb{1}_{N/2} \\ -\mathbb{1}_{N/2} \end{pmatrix}, \lambda_2 = \frac{N(p-q)}{2}.$

Using the definition of the spectral norm, we can bound $\|T_2\|_2$ as

$$\|T_2\|_2 \leq \max_{\|x\|=1} \|(D^{-1} - \bar{D}^{-1})\bar{A}x\|_2$$

$$\leq \max_{\alpha \in \mathbb{R}^2, \|\alpha\|=1} \|(D^{-1} - \bar{D}^{-1})\bar{A}(\alpha_1 w^{(1)} + \alpha_2 w^{(2)})\|_2.$$

Note that when $\|\alpha\|_2 = 1$,

$$\|(D^{-1} - \bar{D}^{-1})\bar{A}(\alpha_1 w^{(1)} + \alpha_2 w^{(2)})\|_2^2 = \sum_{i=1}^{N}\left(\frac{1}{d_i} - \frac{1}{\bar{d}}\right)^2 \left(\sum_{j=1}^{2} \lambda_j \alpha_j w_i^{(j)}\right)^2$$

$$\leq \sum_{i=1}^{N}\left(\frac{1}{d_i} - \frac{1}{\bar{d}}\right)^2 \sum_{j=1}^{2} \lambda_j^2 (w_i^{(j)})^2$$

using Cauchy-Schwarz. Since $|w_i^{(j)}| \leq \frac{1}{\sqrt{N}}$ for all $i, j$, the second summation can be bounded by $\frac{1}{N} \sum_{j=1}^{2} \lambda_j^2$. Overall, the upper bound is now

$$\frac{1}{N} \sum_{i=1}^{N} \frac{(d_i - \bar{d})^2}{(d_i \bar{d})^2} \sum_{j=1}^{2} \lambda_j^2,$$

Under the event of Corollary 1, $d_i \geq C\bar{d}$ for some $C < 1$. Under our setup, we also have $\lambda_1^2 = \bar{d}^2$, $\lambda_2^2 \leq \bar{d}^2$. This means that the upper bound is

$$\frac{1}{C^2 \bar{d}^2 N} \|d - \bar{d}\mathbb{1}_N\|_2^2,$$

where $d$ is the vector of node degrees. It remains to show that $\frac{1}{N}\|d - \bar{d}\mathbb{1}_N\|_2^2 = O(\bar{d})$. To do this, we use a form of Talagrand's concentration inequality, given in Boucheron et al. (2013). Since the function $\frac{1}{\sqrt{N}}\|d - \bar{d}\mathbb{1}_N\|_2 = \frac{1}{\sqrt{N}}\|(A - \bar{d}I_N)\mathbb{1}_N\|_2$ is a convex, 1-Lipschitz function of $A$, Theorem 6.10 from Boucheron et al. (2013) guarantees that for any $t > 0$,

$$\mathbb{P}(\frac{1}{\sqrt{N}}\|d - \bar{d}\mathbb{1}_N\|_2 > \mathbb{E}[\frac{1}{\sqrt{N}}\|d - \bar{d}\mathbb{1}_N\|_2] + t) \leq e^{-t^2/2}.$$

Using Jensen's inequality,

$$\mathbb{E}[\|d - \bar{d}\mathbb{1}_N\|_2] \leq \sqrt{\mathbb{E}[\|d - \bar{d}\mathbb{1}_N\|_2^2]}$$

$$= \sqrt{\sum_{i=1}^{N} \text{Var}(d_i)} = \sqrt{N\text{Var}(d_1)} \leq \sqrt{N\bar{d}}.$$

If $\bar{d} = \omega(\log N)$, we can guarantee that

$$\frac{1}{\sqrt{N}}\|d - \bar{d}\mathbb{1}_N\|_2 \leq C\sqrt{\bar{d}}$$

with probability at least $1 - e^{-(C-1)^2\bar{d}/2} = 1 - O(N^{-r})$ for an appropriate constant $C$. Thus we have shown that with high probability, $T_2 = O(1/\sqrt{\bar{d}})$, which proves the claim. $\qquad \square$

## C.3 Proof of Theorem 1

Fix $r$ and $K$. We desire to bound

$$\frac{1}{N/2} w_2^\top ((D^{-1}A)^k - (\bar{D}^{-1}\bar{A})^k)\mu.$$

By the first property of adjacency matrices in auxiliary results, it suffices to bound

$$\beta \frac{1}{N/2} w_2^\top ((D^{-1}A)^k - (\bar{D}^{-1}\bar{A})^k)w_2.$$

where $\beta = \frac{\mu_1 - \mu_2}{2}$. We will show inductively that there is a $C$ such that for every $k = 1, ..., K$,

$$\|(D^{-1}A)^k - (\bar{D}^{-1}\bar{A})^k\|_2 \leq C/\sqrt{\bar{d}}.$$

If this is true, then Cauchy-Schwarz gives

$$\beta \frac{1}{N/2} w_2^\top ((D^{-1}A)^k - (\bar{D}^{-1}\bar{A})^k)w_2 \leq \beta \frac{1}{N/2}\|w_2\|_2\|(D^{-1}A)^k - (\bar{D}^{-1}\bar{A})^k\|_2\|w_2\|_2$$

$$\leq C/\sqrt{\bar{d}}.$$

By Theorem 3, we have that with probability at least $1 - O(N^{-r})$,

$$\|D^{-1}A - \bar{D}^{-1}\bar{A}\|_2 \leq \frac{C}{\sqrt{\bar{d}}}.$$

So $D^{-1}A = \bar{D}^{-1}\bar{A} + J$ where $\|J\| \leq C/\sqrt{\bar{d}}$. Iterating, we have

$$\|(D^{-1}A)^k - (\bar{D}^{-1}\bar{A})^k\|_2 = \|(D^{-1}A)^{k-1}D^{-1}A - (\bar{D}^{-1}\bar{A})^k\|_2 \tag{1}$$

Inductively, $(D^{-1}A)^{k-1} = (\bar{D}^{-1}\bar{A})^{k-1} + H$ where $\|H\|_2 \leq C/\sqrt{\bar{d}}$. Plugging this in (1), we have

$$\|(D^{-1}A)^{k-1}D^{-1}A - (\bar{D}^{-1}\bar{A})^k\|_2 = \|((\bar{D}^{-1}\bar{A})^{k-1} + H)(\bar{D}^{-1}\bar{A} + J) - (\bar{D}^{-1}\bar{A})^k\|_2\,.$$

Of these terms, $(\bar{D}^{-1}\bar{A})^{k-1}J$ has norm at most $\|J\|_2$, $H(\bar{D}^{-1}\bar{A})$ has norm at most $\|H\|_2$, and $HJ$ has norm at most $C/\bar{d}$.[1] Hence the induction step is complete.

We have thus shown that there is a constant $C(r, K)$ such that with probability at least $1 - N^{-r}$,

$$\left|\frac{1}{N/2}w_2^\top ((D^{-1}A)^k - (\bar{D}^{-1}\bar{A})^k)\mu\right| \leq \frac{C}{\bar{d}}\,.$$

which proves the claim.

By simulation one can verify that indeed $\frac{1}{N/2}w_2^\top(\bar{D}^{-1}\bar{A})^k\mu \approx \left(\frac{p-q}{p+q}\right)^k(\mu_2 - \mu_1)$. Figure 6 presents $\mu_1^{(n)}, \mu_2^{(n)}$ calculated from simulation against predicted values from our theoretical results. The simulation results are averaged over 20 instances generated from CSBM($N = 2000, p = 0.0114, q = 0.0038, \mu_1 = 1, \mu_2 = 1.5, \sigma^2 = 1$).

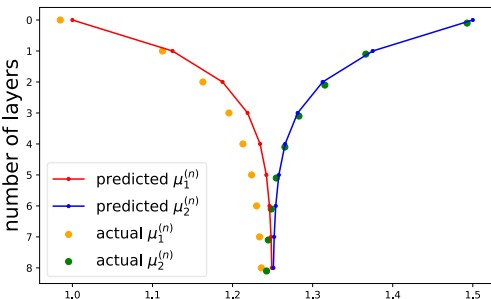

Figure 6: Comparison of the mean estimation in Lemma 2 against simulation results.

## D  PROOF OF LEMMA 3

Fix n and let the element in the $i^{th}$ row and $j^{th}$ column of $(D^{-1}A)^n$ be $p_{ij}^{(n)}$. Consider a fixed node $i$. The variance of the feature for node $i$ after $n$ layers of convolutions is $(\sum_j (p_{ij}^{(n)})^2)\sigma^2$, by the basic property of variance of sum. Since $\sum_j |p_{ij}^{(n)}| = 1$, it follows that $\sum_j (p_{ij}^{(n)})^2 \leq 1$, which is the second inequality.

To show the first inequality, consider the following optimization problem:

$$\min_{p_{ij}^{(n)}, 1 \leq j \leq N} \sum_j (p_{ij}^{(n)})^2$$

$$\text{s.t.} \quad \sum_j p_{ij}^{(n)} = 1,$$

$$p_{ij}^{(n)} \geq 0, \quad 1 \leq j \leq N$$

This part of proof goes by contradiction. Suppose $\exists k, l$ such that $p_{ik}^{(n)} \neq \exists p_{il}^{(n)}$. Fixing all other $p_{ij}^{(n)}, j \neq k, l$, if we average $p_{ik}^{(n)}$ and $p_{il}^{(n)}$, their sum of squares will strictly decrease while not

---

[1]More precisely, the $C$ becomes $C^k$, which is why we restrict the approximation guarantee to constant $K$.

breaking the constraints:

$$2\Big(\frac{p_{ik}^{(n)} + p_{il}^{(n)}}{2}\Big)^2 - ((p_{ik}^{(n)})^2 + (p_{il}^{(n)})^2) = -\frac{1}{2}(p_{ik}^{(n)} - p_{il}^{(n)})^2 < 0 \,.$$

So we obtain a contradiction. Thus to minimize $\sum_j (p_{ij}^{(n)})^2$, $p_{ij}^{(n)} = \frac{1}{N}, 1 \le j \le N$, and the mimimum is $1/N$.

## E    PROOF OF LEMMA 4

The proof relies on the following definition of neighborhood size: in a graph $\mathcal{G}$, we denote by $\Gamma_k(x)$ the set of vertices in $\mathcal{G}$ at distance $k$ from a vertex $x$:

$$\Gamma_k(x) = \{y \in \mathcal{G} : d(x,y) = k\} \,.$$

we define $N_k(x)$ to be the set of vertices within distance $k$ of x:

$$N_k(x) = \bigcup_{i=0}^{k} \Gamma_i(x) \,.$$

To prove the lower bound, we first show an intermediate step that

$$\frac{1}{|N_n|}\sigma^2 \le (\sigma^{(n)})^2 \,.$$

The proof is the same as the one for the first inequality in Lemma 3, except we add in another constraint that for a fixed $i$, the row $p_{i\cdot}$ is $|N_n(i)|$-sparse. This implies that the minimum of $\sum_j (p_{ij}^{(n)})^2$ becomes $1/|N_n(i)|$. The we apply the result on upper bound of neighborhood sizes in Erdős-Rényi graph $\mathcal{G}(N,p)$ (Lemma 2 Graham & Lu (2001)), as it also serves as upper bound of neighborhood sizes in SBM$(N, p, q)$. The result implies that with probability at least $1 - O(1/N)$, we have

$$|N_n| \le \frac{10}{\min\{a, 2\}}(Np)^n \,, \forall 1 \le n \le N \,. \tag{2}$$

We ignore $i$ for $N_n$ because of all nodes are identical in CSBM, so the bound applies for every nodes in the graph.

The proof of upper bound is combinatorial. Corollary 1 states that when $N$ is large, the degree of node $i$ is approximately the expected degree in $\mathcal{G}$, namely, $\mathbb{E}[\text{degree}] = \frac{N}{2}(p + q)$. Since

$$p_{ij}^{(n)} = \sum_{\text{path } P = \{i, v_1, \ldots, v_{n-1}, j\}} \frac{1}{\deg(i)} \frac{1}{\deg(v_1)} \cdots \frac{1}{\deg(v_{n-1})} \,, \tag{3}$$

using the approximation of degrees, we get that

$$p_{ij}^{(n)} = \left(\frac{2}{N(p+q)}\right)^n (\# \text{ of paths } P \text{ of length n between i and j}) \,.$$

Then we use a tree approximation to calculate the number of paths $P$ of length $n$ between $i$ and $j$ by regarding $i$ as the root. Note that

$$\sum_j (p_{ij}^{(n)})^2 = \sum_{k=0}^{\lfloor \frac{n}{2} \rfloor} \sum_{j \in \Gamma_{n-2k}} (p_{ij}^{(n)})^2 \tag{4}$$

and for $j \in \Gamma_{n-2k}$, a deterministic path $P'$ of length $n - 2k$ is needed in order to reach $j$ from $i$. This implies that there are only $k$ steps deviating from $P'$. There are $(n - 2k + 1)^k$ ways of choosing when to deviate. For each specific way of when to deviate, there are approximately $\mathbb{E}[\text{degree}]^k$ ways of choosing the destinations for deviation. Hence in total, for $j \in \Gamma_{n-2k}$, there are $(n - 2k + 1)^k \mathbb{E}[\text{degree}]^k$ path of length $n$ between $i$ and $j$. Thus

$$p_{ij}^{(n)} = (n - 2k + 1)^k \left(\frac{2}{N(p+q)}\right)^{n-k} \,. \tag{5}$$

Plug in (5) into (4), we get that

$$\sum_j (p_{ij}^{(n)})^2 = \sum_{k=0}^{\lfloor \frac{n}{2} \rfloor} |\Gamma_{n-2k}|(n-2k+1)^{2k} \left( \frac{2}{N(p+q)} \right)^{2n-2k} \tag{6}$$

$$\leq \sum_{k=0}^{\lfloor \frac{n}{2} \rfloor} \frac{9}{\min\{a,2\}} (n-2k+1)^{2k} (Np)^{n-2k} \left( \frac{2}{N(p+q)} \right)^{2n-2k} \tag{7}$$

Again, (7) follows from using the upper bound on $|\Gamma_{n-2k}|$ Graham & Lu (2001) such that with probability at least $1 - O(1/N)$,

$$|\Gamma_{n-2k}| \leq \frac{9}{\min\{a,2\}} (Np)^{n-2k}, \forall 1 \leq k \leq \lfloor \frac{n}{2} \rfloor.$$

Combining with Lemma 3, we obtain the final result.

Figure 7 presents variance calculated from simulation against predicted upper and lower bounds from our theoretical results. The simulation results are averaged over 1000 instances generated from CSBM$(N = 2000, p = 0.0114, q = 0.0038, \mu_1 = 1, \mu_2 = 1.5, \sigma^2 = 1)$.

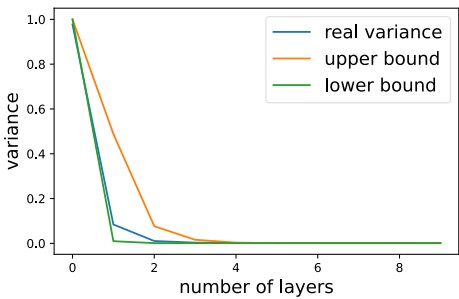

Figure 7: Comparison of the bounds on variance in Theorem 2 against simulation results.

## F  PROOF OF THEOREM 2

When we fix $K \in \mathbb{N}$, only the upper bound in Theorem 2 will change. Note that now the upper bound in (7) can be written as

$$\sum_{k=0}^{\lfloor \frac{n}{2} \rfloor} \frac{9}{\min\{a,2\}} (n-2k+1)^{2k} \left( \frac{p+q}{2p} \right)^{2k} \left( \frac{2p}{p+q} \right)^n \left( \frac{2}{N(p+q)} \right)^n$$

$$\leq \frac{C}{\min\{a,2\}} \left( \sum_{k=0}^{C} \left( \frac{p+q}{2p} \right)^{2k} \right) \left( \frac{2}{N(p+q)} \right)^n$$

$$\leq \frac{C}{\min\{a,2\}} \left( \frac{2}{N(p+q)} \right)^n.$$

## G  PROOF OF PROPOSITION 1

Let the node representation vector of node $v$ after $n$ graph convolutions be $h_v^{(n)}$. The Bayes error rate could be written as $\frac{1}{2}(\mathbb{P}[h_v^{(n)} > \mathcal{D}|v \in \mathcal{C}_1] + \mathbb{P}[h_v^{(n)} \leq \mathcal{D}|v \in \mathcal{C}_2])$. For $d \in \mathbb{N}$, due to the symmetry of our setup, one can easily see that the optimal linear decision boundary is the hyperplane $\sum_{j=1}^{d} x_j = \frac{d}{2}(\mu_1 + \mu_2)$. Then for $v \in \mathcal{C}_1$, $\sum_{j=1}^{d} (h_v^{(n)})_j \sim \mathcal{N}(d\mu_1^{(n)}, d(\sigma^{(n)})^2)$ and for $v \in \mathcal{C}_2$,

$\sum_{j=1}^{d}(h_v^{(n)})_j \sim \mathcal{N}(d\mu_2^{(n)}, d(\sigma^{(n)})^2)$. Thus the Bayes error rate can be written as

$$\frac{1}{2}(\mathbb{P}[\sum_{j=1}^{d}(h_v^{(n)})_j > \mathcal{D}|v \in \mathcal{C}_1] + \mathbb{P}[\sum_{j=1}^{d}(h_v^{(n)})_j \leq \mathcal{D}|v \in \mathcal{C}_2])$$

$$= \frac{1}{2}\left(1 - \Phi\left(\frac{\frac{d}{2}(\mu_1 + \mu_2) - d\mu_1^{(n)}}{\sqrt{d}\sigma^{(n)}}\right)\right) + \frac{1}{2}\left(\Phi\left(\frac{\frac{d}{2}(\mu_1 + \mu_2) - d\mu_2^{(n)}}{\sqrt{d}\sigma^{(n)}}\right)\right)$$

$$= 1 - \Phi\left(\frac{\frac{d}{2}(\mu_1 + \mu_2) - d\mu_1^{(n)}}{\sqrt{d}\sigma^{(n)}}\right).$$

The last equality follows from the fact that $\frac{d}{2}(\mu_1 + \mu_2) - d\mu_1^{(n)} = -(\frac{d}{2}(\mu_1 + \mu_2) - d\mu_2^{(n)})$.

## H HOW TO USE THE Z-SCORE TO CHOOSE THE NUMBER OF LAYERS

The bounds of the z-score with respect to the number of layers, $z_{\text{lower}}^{(n)}$ and $z_{\text{upper}}^{(n)}$ allow us to calculate bounds for $n^\star$ and $n_0$ under different scenarios. Specifically,

1. $\forall n \in \mathbb{N}, z_{\text{upper}}^{(n)} < z^{(0)} = (\mu_2 - \mu_1)/\sigma$, then $n^\star = n_0 = 0$, meaning that no graph convolution should be applied.

2. $|\{n \in \mathbb{N} : z_{\text{upper}}^{(n)} \geq z^{(0)}\}| > 0$, and

   (a) $\forall n \in \mathbb{N}, z_{\text{lower}}^{(n)} < z^{(0)}$, then $0 \leq n_0 \leq \min\{n \in \mathbb{N} : z_{\text{upper}}^{(n)} \leq z^{(0)}\}$, which means that the number of graph convolutions should not exceed the upper bound of $n_0$, or otherwise one gets worse performance than having no graph convolution. Note that in this case, since $n^\star \leq n_0$, we can only conclude that

   $$0 \leq n^\star \leq \min\{n \in \mathbb{N} : z_{\text{upper}}^{(n)} \leq z^{(0)}\}.$$

   (b) $|\{n \in \mathbb{N} : z_{\text{lower}}^{(n)} \geq z^{(0)}\}| > 0$, then $0 \leq n_0 \leq \min\{n \in \mathbb{N} : z_{\text{upper}}^{(n)} \leq z^{(0)}\}$, and let $\arg\max_n z_{\text{lower}}^{(n)} = n_{\text{floor}}^\star$,

   $$\max\left\{n \leq n_{\text{floor}}^\star : z_{\text{upper}}^{(n)} \leq z_{\text{lower}}^{(n_{\text{floor}}^\star)}\right\} \leq n^\star \leq \min\left\{n \geq n_{\text{floor}}^\star : z_{\text{upper}}^{(n)} \leq z_{\text{lower}}^{(n_{\text{floor}}^\star)}\right\},$$

   meaning that the number of layers one should apply for optimal node classification performance is more than the lower bound of $n^\star$, and less than the upper bound of $n^\star$.

## I PROOFS OF PROPOSITION 2-5

### I.1 PROOF OF PROPOSITION 2

Since the spectral radius of $D^{-1}A$ is 1,

$$\alpha(Id - (1-\alpha)(D^{-1}A))^{-1} = \alpha\sum_{k=0}^{\infty}(1-\alpha)^k(D^{-1}A)^k.$$

Apply Lemma 2, we get that $\mu_2^{\text{PPNP}} - \mu_1^{\text{PPNP}} \approx \frac{p+q}{p+\frac{2-\alpha}{\alpha}q}(\mu_2 - \mu_1)$.

To bound the approximation error, similar to the proof of the concentration bound in Theorem 1, it suffices to bound

$$\frac{\mu_1 - \mu_2}{N}w_2^\top(\sum_{k=0}^{\infty}\alpha(1-\alpha)^k((D^{-1}A)^k - (\bar{D}^{-1}\bar{A})^k))w_2 = \frac{\mu_1 - \mu_2}{N}w_2^\top(T_K + T_{K+1,\infty})w_2,$$

where $T_K = \sum_{k=0}^{K}\alpha(1-\alpha)^k((D^{-1}A)^k - (\bar{D}^{-1}\bar{A})^k)$, $T_{K+1,\infty} = \sum_{k=K+1}^{\infty}\alpha(1-\alpha)^k((D^{-1}A)^k - (\bar{D}^{-1}\bar{A})^k)$, and $K \in \mathbb{N}$ up to our own choice.

**Bounding $T_K$:** Apply Theorem 1, fix $r > 0$, there exists a constant $C(r, K, \alpha)$ such that with probability $1 - O(N^{-r})$,

$$\|T_K\|_2 \leq \frac{C}{\sqrt{\bar{d}}}.$$

**Bounding $T_{K+1,\infty}$:** We will show upper bound for $(D^{-1}A)^k - (\bar{D}^{-1}\bar{A})^k$ that applies for all $k \in \mathbb{N}$. Note that for every $k \in \mathbb{N}$,

$$(D^{-1}A)^k = D^{-1/2}(D^{-1/2}AD^{-1/2})^k D^{1/2} = D^{-1/2}(V\Lambda^k V^\top)D^{1/2},$$

where $D^{-1/2}AD^{-1/2} = V\Lambda V^\top$ is the eigenvalue decomposition. Then

$$\|(D^{-1}A)^k - (\bar{D}^{-1}\bar{A})^k)\|_2 \leq \|(D^{-1}A)^k\|_2 + \|(\bar{D}^{-1}\bar{A})^k\|_2 = \|(D^{-1}A)^k\|_2 + 1$$
$$\leq \|D^{-1/2}\|_2 \|(D^{-1/2}AD^{-1/2})^k\|_2 \|D^{-1/2}\|_2 + 1.$$

Since $\|(D^{-1/2}AD^{-1/2})^k\|_2 = 1$ and by Corollary 1, with probability at least $1 - N^{-r}$,

$$\|D^{1/2}\|_2 \leq \sqrt{3\bar{d}/2}, \|D^{-1/2}\|_2 \leq \sqrt{2/\bar{d}},$$

the previous inequality becomes $\|(D^{-1}A)^k - (\bar{D}^{-1}\bar{A})^k)\|_2 \leq \sqrt{3} + 1$. Hence

$$\|T_{K+1,\infty}\|_2 \leq (1-\alpha)^{K+1}.$$

Combining the two results, we prove the claim.

## I.2    PROOF OF PROPOSITION 3

The claim is a direct corollary of Theorem 1.

## I.3    PROOF OF PROPOSITION 4

The covariance matrix $\Sigma^{\text{PPNP}}$ of $h^{\text{PPNP}}$ could be written as

$$\Sigma^{\text{PPNP}} = \alpha^2 (\sum_{k=0}^{\infty} (1-\alpha)^k (D^{-1}A)^k)(\sum_{l=0}^{\infty}(1-\alpha)^l(D^{-1}A)^l)^\top \sigma^2.$$

Note that the variance of node $i$ equals $\alpha^2 \sum_{k,l=0}^{\infty}(1-\alpha)^{k+l}(D^{-1}A)_{i\cdot}^k((D^{-1}A)^l)_{i\cdot}^\top$, where $i\cdot$ refers row $i$ of a matrix. Then by Cauchy-Schwarz Theorem,

$$(D^{-1}A)_{i\cdot}^k((D^{-1}A)^l)_{i\cdot}^\top \leq \|(D^{-1}A)_{i\cdot}^k\|\|((D^{-1}A)^l)_{i\cdot}\|$$
$$\leq \sqrt{(\sigma^{(k)})^2(\sigma^{(l)})^2/\sigma^2}, \text{ for all } 1 \leq k,l \leq N.$$

Moreover, by Lemma 3, $(\sigma^{(k)})^2 \leq \sigma^2$. Due to the identity of each node $i$, we get that with probability $1 - O(1/N)$, for all $1 \leq K \leq N$,

$$(\sigma^{\text{PPNP}})^2 \leq \alpha^2 \left( \sum_{k=0}^{K}(1-\alpha)^k \sqrt{(\sigma^{(k)})_{\text{upper}}^2} + \sum_{k=K+1}^{\infty}(1-\alpha)^k \sigma \right)^2$$
$$\leq \alpha^2 \left( \sum_{k=0}^{K}(1-\alpha)^k \sqrt{(\sigma^{(k)})_{\text{upper}}^2} + \frac{(1-\alpha)^{K+1}}{\alpha}\sigma \right)^2.$$

For the lower bound, note that with probability $1 - O(1/N)$,

$$(\sigma^{\text{PPNP}})^2 \geq \alpha^2 \left( \sum_{k=0}^{N}(1-\alpha)^{2k}\frac{1}{N_k} + \sum_{k=N+1}^{\infty}(1-\alpha)^{2k}\frac{1}{N} \right)\sigma^2,$$

where $N_k$ is the size of $k$-hop neighborhood. Then

$$(\sigma^{\text{PPNP}})^2 \geq \alpha^2 \left( \sum_{k=0}^{N} (1-\alpha)^{2k} \frac{\min\{a, 2\}}{10} \frac{1}{(Np)^k} \right) \sigma^2$$

$$\geq \alpha^2 \frac{\min\{a, 2\}}{10} \frac{(Np)^{N+1} - (1-\alpha)^{2N+2}}{(Np)^N (Np - (1-\alpha)^2)} \sigma^2$$

$$\geq \alpha^2 \frac{\min\{a, 2\}}{10} \sigma^2 .$$

It is easy to see that Lemma 3 applies to any message-passing scheme which could be regarded as a random walk on the graph. Combining with Lemma 3, we get the final result.

### I.4 PROOF OF PROPOSITION 5

Since

$$h^{\text{APPNP}(n)} = \left( \alpha \left( \sum_{k=0}^{n-1} (1-\alpha)^k (D^{-1}A)^k \right) + (1-\alpha)^n (D^{-1}A)^n \right) X$$

Through the same calculation as for the upper bound in the proof of Proposition 2, we get that with probability $1 - O(1/N)$,

$$(\sigma^{\text{APPNP}(n)})^2 \leq \left( \alpha \left( \sum_{k=0}^{n-1} (1-\alpha)^k \sqrt{(\sigma^{(k)})^2_{\text{upper}}} \right) + (1-\alpha)^n \sqrt{(\sigma^{(n)})^2_{\text{upper}}} \right)^2 .$$

For the lower bound, through the same calculation as for the upper bound in the proof of Proposition 2, we get that with probability $1 - O(1/N)$,

$$(\sigma^{\text{APPNP}(n)})^2 \geq \alpha^2 \sum_{k=0}^{n-1} (1-\alpha)^{2k} (\sigma^{(k)})^2 + (1-\alpha)^{2n} (\sigma^{(n)})^2$$

$$\geq \alpha^2 \frac{\min\{a, 2\}}{10} \left( \sum_{k=0}^{n-1} (1-\alpha)^{2k} \frac{1}{(Np)^k} \right) \sigma^2 + \frac{\min\{a, 2\}}{10} (1-\alpha)^{2n} \frac{1}{(Np)^n} \sigma^2$$

$$\geq \frac{\min\{a, 2\}}{10} \left( \alpha^2 + \frac{(1-\alpha)^{2n}}{(Np)^n} \right) \sigma^2 .$$

Combining with Lemma 3, we get the final result.

## J EXPERIMENTS

Here we provide more details on the models that we use in Section 5. In all cases we use the Adam optimizer and tune some hyperparameters for better performance. The hyperparameters used are summarized as follows.

| Data | final linear classifier | weights in GNN layer | learning rate (width) | iterations (width) |
|------|------------------------|---------------------|----------------------|-------------------|
| synthetic | 1 layer | no | 0.01 | 8000 |
| | | yes | 0.01(1,4,16)/0.001(64,256) | 8000(1,4,16)/10000(64)/50000(256) |
| Cora | 3 layer with 32 hidden channels | no | 0.001 | 150 |
| | | yes | 0.001 | 200 |
| CiteSeer | 3 layer with 16 hidden channels | no | 0.001 | 100 |
| | | yes | 0.001 | 100 |
| PubMed | 3 layer with 32 hidden channels | no | 0.001 | 500 |
| | | yes | 0.001 | 500 |

We empirically find that after adding in weights in each GNN layer, it takes much longer to train the model for one iteration, and the time increases when the depth or the width increases (Figure 8). Since for some combinations, it takes more than 200,000 iterations for the validation accuracy to finally increase, for each case, we only train for a reasonable amount of iterations.

All models were implemented with PyTorch (Paszke et al., 2019) and PyTorch Geometric (Fey & Lenssen, 2019).

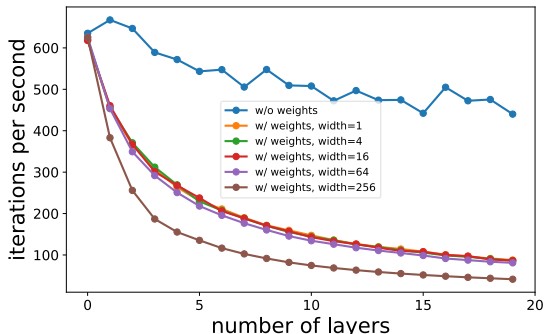

Figure 8: Iterations per second for each model.

## K  ADDITIONAL RESULTS

### K.1  EFFECT OF NONLINEARITY ON CLASSIFICATION PERFORMANCE

In section 3, we consider the case of a simplified linear GNN. What would happen if we add nonlinearity after linear graph convolutions? Here, we consider the case of a GNN with a ReLU activation function added after $n$ linear graph convolutions, i.e. $h^{(n)\text{ReLU}} = \text{ReLU}((D^{-1}A)^n X)$. We show that adding such nonlinearity does not improve the classification performance.

**Proposition 6.** *Applying a ReLU activation function after n linear graph convolutions does not decrease the Bayes error rate, i.e. Bayes error rate based on $h^{(n)ReLU} \geq$ Bayes error rate based on $h^{(n)}$, and equality holds if $\mu_1 \geq -\mu_2$.*

*Proof.* If is known that if $x$ follows a Gaussian distribution, then $\text{ReLU}(x)$ follows a Rectified Gaussian distribution. Following the definition of the Bayes optimal classifier, we present a geometric proof in Figure 9 (see next page, top), where the dark blue bar denotes the location of $0$ and the red bar denotes the decision boundary $\mathcal{D}$ of the Bayes optimal classifier, and the light blue area denotes the overlapping area $S$, which is twice the Bayes error rate.

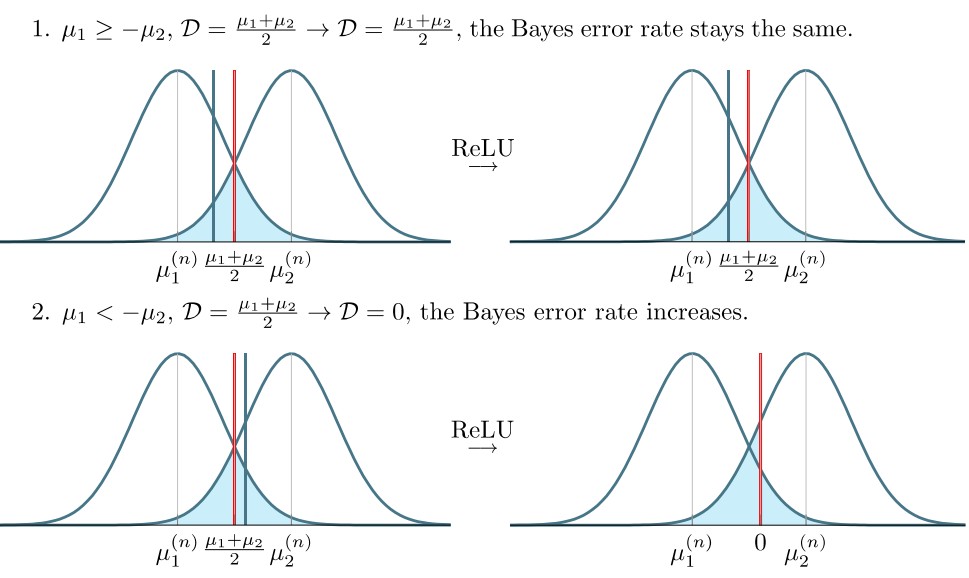

Figure 9: A geometric proof of Proposition 6.

$\square$

## K.2 Exact limit of variance $(\sigma^{(n)})^2$ as $n \to \infty$

**Proposition 7.** *Given a graph $\mathcal{G}$ with adjacency matrix $A$, let its degree vector be $d = A\mathbb{1}_N$, where $\mathbb{1}_N$ is the all-one vector of length $N$. If $\mathcal{G}$ is connected and non-bipartite, the variance of each node $i$, denoted as $(\sigma_i^{(n)})^2$, converges asymptotically to $\frac{\|d\|_2^2}{\|d\|_1^2}$, i.e.*

$$(\sigma_i^{(n)})^2 \xrightarrow{n \to \infty} \frac{\|d\|_2^2}{\|d\|_1^2}.$$

*Then $\frac{\|d\|_2^2}{\|d\|_1^2} \geq \frac{1}{N}$, and the equality holds if and only if $\mathcal{G}$ is regular.*

*Proof.* Let $e_i$ denotes the standard basis unit vector with the $i^{th}$ entry equals 1, and all other entries equal 0. Since $\mathcal{G}$ is connected and non-bipartite, the random walk represented by $P = D^{-1}A$ is ergodic, meaning that

$$e_i^\top P^{(n)} \xrightarrow{n \to \infty} \pi,$$

where $\pi$ is the stationary distribution of this random walk with $\pi_i = \frac{d_i}{\|d\|_1}$. Then since norms are continuous functions, we conclude that

$$(\sigma_i^{(n)})^2 = \sum_j (p_{ij}^{(n)})^2 = \|e_i^\top P^{(n)}\|_2^2 \xrightarrow{n \to \infty} \|\pi\|_2^2 = \frac{\|d\|_2^2}{\|d\|_1^2}.$$

By Lemma 3, it follows that $\frac{\|d\|_2^2}{\|d\|_1^2} \geq \frac{1}{N}$. The unique minimizer of $\|\pi\|_2^2$ subject to $\|\pi\|_1 = 1$ is $\pi = \frac{1}{N}\mathbb{1}_N$. This means that $\mathcal{G}$ must be regular to achieve the lower bound asymptotically. $\square$

Under Assumption 1, the graph generated by our CSBM is almost surely connected. Here, we remain to show that with high probability, the graph will also be non-bipartite.

**Proposition 8.** *With probability at least $1 - O(1/(Np)^3)$, a graph $\mathcal{G}$ generated from CSBM($N$, $p$, $q$, $\mu_1$, $\mu_2$, $\sigma^2$) contains a triangle, which implies that it is non-bipartite.*

*Proof.* The proof goes by the classic probabilistic method. Let $T_\Delta = \sum_i^{\binom{N}{3}} \mathbb{1}_{\tau_i}$ denotes the number of triangles in $\mathcal{G}$, where $\mathbb{1}_{\tau_i}$ equals 1 if potential triangle $\tau_i$ exists and 0 otherwise. Then by second moment method,

$$\mathbb{P}[T_\Delta = 0] \leq \frac{\text{Var}(T_\Delta)}{(\mathbb{E}[T_\Delta])^2} = \frac{1}{\mathbb{E}[T_\Delta])} + \frac{\sum_{i \neq j} \mathbb{E}[\mathbb{1}_{\tau_i}\mathbb{1}_{\tau_j}] - (\mathbb{E}[T_\Delta])^2}{(\mathbb{E}[T_\Delta])^2}.$$

Since $\mathbb{E}[T_\Delta] = O(Np)$, $\sum_{i \neq j} \mathbb{E}[\mathbb{1}_{\tau_i}\mathbb{1}_{\tau_j}] = (1 + O(1/N))(\mathbb{E}[T_\Delta])^2$, we get that

$$\mathbb{P}[T_\Delta = 0] \leq O(1/(Np)^3) + O(1/N) \leq O(1/(Np)^3).$$

Hence $\mathbb{P}[\mathcal{G} \text{ is non-bipartite}] \geq \mathbb{P}[T_\Delta \geq 1] \geq 1 - O(1/(Np)^3)$. $\square$

## K.3 Symmetric Graph Convolution $D^{-1/2}AD^{-1/2}$

**Proposition 9.** *When using symmetric message-passing convolution $D^{-1/2}AD^{-1/2}$ instead, the variance $(\sigma^{(n)})^2$ is non-increasing with respect to the number of convolutional layers $n$. i.e.*

$$(\sigma^{(n+1)})^2 \leq (\sigma^{(n)})^2, n \in \mathbb{N} \cup \{0\}.$$

*Proof.* We want to calculate the diagonal entries of the covariance matrix $\Sigma^{(n)}$ of $(D^{-1/2}AD^{-1/2})^n X$, where the covariance matrix of $X$ is $\sigma^2 I_N$. Hence

$$\Sigma^{(n)} = (D^{-1/2}AD^{-1/2})^n \big((D^{-1/2}AD^{-1/2})^n\big)^\top.$$

Since $D^{-1/2}AD^{-1/2}$ is symmetric, let its eigendecomposition be $V\Lambda V^\top$ and we could rewrite

$$\Sigma^{(n)} = (V\Lambda^n V^\top)(V\Lambda^n V^\top) = V\Lambda^{2n} V^\top.$$

Notice that the closed form of the diagonal entries is

$$\text{diag}(\Sigma^{(n)}) = \sum_{i=1}^{N} \lambda_i^{2n} |v|^2 \,.$$

Since for all $1 \leq i \leq N$, $|\lambda_i| \leq 1$, we obtain monotonicity of each entry of $\text{diag}(\Sigma^{(n)})$, i.e. variance of each node. $\quad\square$

Although the proposition does not always hold for random walk message-passing convolution $D^{-1}A$ as one can construct specific counterexamples (Appendix K.4), in practice, variances are observed to be decreasing with respect with the number of layers. Moreover, we empirically observe that variance goes down more than the variance using symmetric message-passing convolutions. Figure 10 presents visualization of node representations comparing the change of variance with respect to the number of layers using random walk convolution and symmetric message-passing convolution. The data is generated from CSBM($N = 2000, p = 0.0114, q = 0.0038, \mu_1 = 1, \mu_2 = 1.5, \sigma^2 = 1$).

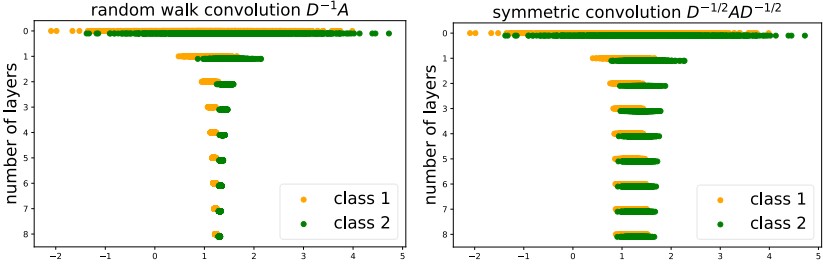

Figure 10: The change of variance with respect to the number of layers using random walk convolution $D^{-1}A$ and symmetric message-passing convolution $D^{-1/2}AD^{-1/2}$.

### K.4 COUNTEREXAMPLES

Here, we construct a specific example where the variance $(\sigma^{(n)})^2$ is not non-increasing with respect to the number of layers $n$ (Figure 11A). We remark that such a non-monotone nature of change in variance is not caused by the bipartiteness of the graph, as a cycle graph with even number of nodes is also bipartite, but does not exhibit such phenomenon (Figure 11B). We conjecture the increase in variance is rather caused by the tree-like structure.

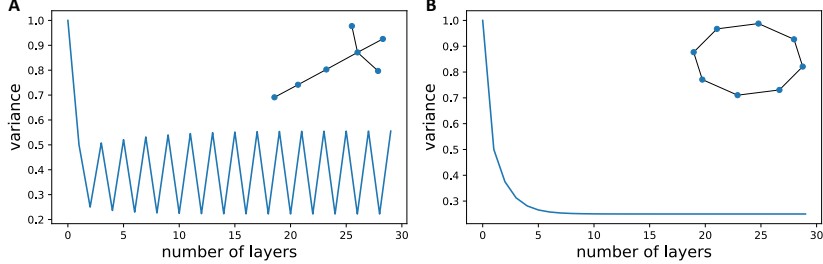

Figure 11: Counterexamples.

### K.5 THE MIXING AND DENOISING EFFECTS IN PRACTICE

In this section, we measure the mixing and denoising effects of graph convolutions identified by our theoretical results in practice, and show that the same tradeoff between the two counteracting effects exists for real-world graphs. For the mixing effect, we measure the pairwise $L_2$ distances between the means of different classes, and for the denoising effect, we measure the within-class variances, both respect to the number of layers. Figure 12 gives a visualization of both metrics for

all classes on Cora, CiteSeer and PubMed. We observe that similar to the synthetic CSBM data, adding graph convolutions increases both the mixing effect (homogenizing node representations in different classes, measured by the inter-class distances) and the denoising effect (homogenizing node representations in the same class, measured by the within-class distances). In addition, the beneficial denoising effect clearly reaches saturation just after a small number of layers, as predicted by our theory.

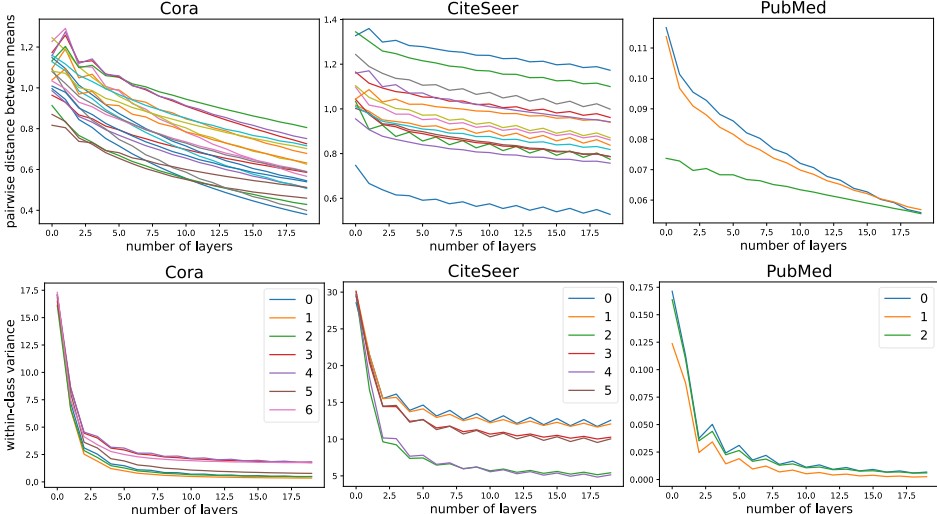

Figure 12: The existence of the mixing (top row) and denoising effects (bottom row) of graph convolutions in practice. Adding graph convolutions increases both effects and the beneficial denoising effect clearly reaches saturation just after a small number of layers, as predicted by our theory in Section 3.

