# OpenReview forum: "A Non-Asymptotic Analysis of Oversmoothing in Graph Neural Networks"
_ICLR.cc/2023/Conference — ICLR 2023 poster_

### Official Review · Reviewer_K5CA · 2022-10-25

**Confidence:** 2
**Correctness:** 4
**Technical Novelty And Significance:** 4
**Empirical Novelty And Significance:** 4
**Recommendation:** 8

**Clarity, Quality, Novelty And Reproducibility:**

The paper is well-written and easy to follow. As far as I can tell, the results are new and different in scope from existing theory on oversmoothing.

**Strength And Weaknesses:**

General comments
-

- In the experiments section, your test accuracies for Cora, Citeseer, and Pubmed with weights appear much lower than in other papers. Is there a reason for this based on your experimental setup?

Strengths
-

- The paper improves upon theoretical results on oversmoothing from past work, which proved crude bounds using Laplacian eigenvalues. In contrast, this paper finds bounds for shallow networks which are a more realistic regime for node classification.
- Contextual SBMs are a good choice of framework for the theory that capture causes of oversmoothing in real-world graphs.

Weaknesses:
-

- The node feature vectors are assumed to be 1-dimensional; is this strictly necessary to prove the results of the paper?
- The Bayes optimal classifier is a simplification (as noted by the authors) and may not be applicable to the training of GNNs.
- Performance does not seem to degrade much for Cora, Citeseer, and Pubmed in the case without weights even for very deep networks. This is in sharp contrast to the CSBM data, indicating that there are different factors contributing to the synthetic data.

**Summary Of The Paper:**

This paper provides a theoretical analysis of oversmoothing in linear GCNs in an SBM setting where the features and labels are assumed to be Gaussian. It provides bounds for shallow GNNs rather than convergence rates in the infinite-layer limit. The theoretical results are supported with empirical data showing the effect of oversmoothing for synthetic and real-world datasets as the number of layers increases.

**Summary Of The Review:**

The paper answers questions about the extent of oversmoothing in GNNs, and for a specific model exhibiting community structure, shows that oversmoothing occurs after a relatively small number of layers which can be explicitly predicted. While several simplifying assumptions are made, enough structure is imposed on the synthetic graphs for the results to be meaningful.

---

> ### Author Response · Authors · 2022-11-18
> **Response to Reviewer K5CA**
>
> We appreciate your thoughtful comments and positive assessment of our work. After carefully reviewing your feedback, below we provide answers to the comments you raised.
>
> **Q1 Your test accuracies for Cora, Citeseer, and Pubmed with weights appear much lower than in other papers. Is there a reason for that?**
>
> We follow the same experimental setup as [1] and we observe that once we increase the width to $512$ (brown curve), the test accuracies in our experiment are on par with the results provided in [1] (Table 2: $\approx$ 85% for Cora, $\approx$ 73% for CiteSeer and $\approx$ 88% for PubMed).
>
> It is possible that the hyperparameters are not fully optimized for some of the narrower models, as we observe (and it is known) that these models are much harder to tune/train [2].
>
> **Q2 Is node features being 1-dimensional strictly necessary to prove the results of the paper?**
>
> Thank you for the question. Features being one-dimensional is not fundamental to our results. We show in Proposition 2 the case of $d$-dimensional features under the assumption that each dimension is independent. The original statement of the result was a bit unclear. We have updated the manuscript to make Proposition 2 more precise.
>
> **Q3 The Bayes optimal classifier is a simplification (as noted by the authors) and may not be applicable to the training of GNNs.**
>
> Indeed, the Bayes optimal classifier is a simplification. However, the experiments in our work show that the theoretical prediction based on the Bayes optimal classifier is very close to the empirical performance of a classifier trained in a standard semi-supervised setting (Figure 2A&4A). So the Bayes optimal classifier is an adequate indicator for the actual GNN classification performance in our case.
>
> **Q4 Performance does not seem to degrade much for Cora, Citeseer, and Pubmed in the case without weights even for very deep networks. This is in sharp contrast to the CSBM data.**
>
> This is a good point. The CSBM is not a perfect model for real-world graphs, in aspects such as degree heterogeneity (as noted in the paper). Nonetheless, while it does not capture everything in real graphs, we believe the CSBM is capturing the most relevant graph features to the problem that we are interested in so that we can identify the core mechanism behind oversmoothing, which is the main focus of this work. We have included a set of experiments showing that a similar tradeoff between the two effects of graph convolutions exists in real-world graphs in Appendix L.4 of the updated manuscript.
>
> We appreciate your questions and comments very much. Please let us know for any further questions.
>
> [1]Huang et al. Combining label propagation and simple models outperforms graph neural networks. ICLR, 2021.
>
> [2]Du and Hu. Width provably matters in optimization for deep linear neural
> networks. ICML, 2019.

---

### Official Review · Reviewer_KQXD · 2022-10-26

**Confidence:** 3
**Correctness:** 4
**Technical Novelty And Significance:** 3
**Empirical Novelty And Significance:** 2
**Recommendation:** 6

**Clarity, Quality, Novelty And Reproducibility:**

Overall, this paper is clear and helpful work for understanding oversmoothing in GNNS. To my best knowledge, it is a novel work that shows a new theoretical view on the oversmoothing phenomenon.

**Strength And Weaknesses:**


1. This work provides a non-asymptotic analysis formulation of oversmoothing, closing the gap between finite-depth GNNs and the previous asymptotic analyses. The authors provide an explanation theoretically for why the optimal number of GNN layers for node classification is relatively small and gives a sweet spot for the required number of layers.
2. The distinguishment of the mixing effect and denoising effect is helpful to understand counteracting effects in neural message passing.
3. Experimentally, they observe that the difficulty of optimizing weight in deep GNNs aggravates oversmoothing. It makes sense and might be inspiring in further message passing designs.
4. It seems that all the analysis relies on the assumption that the graph is generated as $\mathcal(A,X)\sim\mathrm{CSBM}(N, p, q, \mu_1, \mu_2, \sigma^2.)$ The authors point that their analysis cannot capture graph characteristics like degree heterogeneity. But except it, whether the analysis formulation can be applied to general feature distributions?

**Questions**

1. Could you please state more about the advantages of non-asymptotic analysis? To my knowledge, oversmoothing is associated with the Dirichlet energy decay rate in the previous works. In this work, the domination of the mixing effect is considered as the reason of it. Is there any connection between the two demonstrations (decay rate and domination)?
2. The `sweet spot’ given in the paper is indeed a lovely choice for avoiding oversmoothing. I am curious about whether it is a sweet spot for the whole learning process. For example, is it possible that the sweet spot can be too short for some long-range problems?
3. Could you give a notation instruction for $\omega(logN/N)$ in Page 3 and $\Omega(logN/N)$ in Page 4? And could you explain more about Figure 2? There are few illustrations, especially on graphs B and C.
4. In the abstract, it says ‘PPR-based architectures still achieve their best performance at a shallow depth and are outperformed by the graph convolution approach on certain graphs.’ And Section 4, says ‘This drawback would be especially notable at a shallow depth, where the denoising effect is supposed to dominate the mixing effect. PPNP/APPNP would perform worse than the baseline GNN on these graphs in terms of the optimal classification performance.’ How does PPNP perform at a shallow depth after all?



**Summary Of The Paper:**

This paper characterizes the mechanism of oversmoothing by a non-asymptotic analysis. It distinguishes undesirable mixing effect and the desirable denoising effect on graph convolutions and gives an estimation of when the mixing effect dominates the denoising effect, by quantifying both the effects on random graphs sampled from Contextual Stochastic Block Model (CSBM). On the basis of this analysis, it explains the oversmoothing phenomenon at a relatively shallow depth. To show that the framework can be applied to other message-passing schemes, specifically, the authors analyze the performance of Personalized Propagation of Neural Predictions (PPNP) and Approximate PPNP (APPNP). Their results suggest that PPR-based architectures mitigate oversmoothing at deep layers. Additionally, some numerical experiments support the theoretical results and indicate that the difficulty of optimizing parameters exacerbates oversmoothing.

**Summary Of The Review:**

This paper provides a novel perspective to understand oversmoothing and a non-asymptotic framework to analyze it. It gives a series of theoretical results in both the main body and the appendix with detailed proofs. Their experiments support their main results and imply the detriment of the difficulty of optimizing deep GNN models.

---

> ### Author Response · Authors · 2022-11-18
> **Response to Reviewer KQXD**
>
> Thank you for your constructive feedback and positive assessment of our work. Below, we provide individual responses to your questions.
>
> **Q1 Can the analysis be applied if the node feature is not Gaussian-distributed?**
>
> This is a good question. The Gaussian assumption can indeed be relaxed. For a general feature distribution with a bounded second moment, after one or more graph convolutions, the node representation will become approximately Gaussian by the central limit theorem, and the rest of the argument would apply under additional concentration bounds. Here, we adopt the Gaussian assumption in the paper for simplicity and clarity in the exposition of the main results.
>
> **Q2 The advantages of non-asymptotic analysis?**
>
> Thanks for the question.
> - The non-asymptotic analysis allows us to explain why oversmoothing happens in GNNs with a *small, finite depth*, as opposed to only with a very large depth predicted by the asymptotic theory. It has been empirically observed that oversmoothing starts to kick in for as few as 2-4 layers [1], which cannot be fully explained by the existing asymptotic analyses [2,3], as they only show that oversmoothing is inevitable when the number of layers goes to infinity. In this work, we develop a new theoretical analysis framework for studying the behavior of finite-depth GNNs and prove the first quantitative theory of oversmoothing in GNNs with a small, finite depth.
> - Moreover, *only* with a non-asymptotic analysis are we able to show that methods proposed earlier for solving oversmoothing such as APPNP (or equivalently, initial residual connections in [4]) provably don't work in some instances. We provide a rigorous explanation for why they cannot fundamentally solve oversmoothing, thus paving the way for a better usage of such models. Since these methods were built on the asymptotic understanding, their potential disadvantages have been overlooked because people don't fully understand why oversmoothing happens in finite-depth GNNs.
>
> **Q3 Any connection with the Dirichlet energy framework?**
>
> The Dirichlet energy is a measure of non-smoothness of graph signals. By establishing a connection between graph convolutions and gradient flow with respect to the Dirichlet energy, prior works such as [5] provide an alternative argument for the smoothing effect of graph convolutions. However, since the definition of the Dirichlet energy is agnostic to the nodes' classes, such analyses do not distinguish between smoothing across different classes (undesirable) and smoothing within the same class (desirable). Hence the energy decay is unable to characterize the tradeoff between the two effects, which is essential to explaining oversmoothing as well as the benefits of graph convolutions at a finite depth. i.e. the Dirichlet energy framework could only provide an asymptotic theory of oversmoothing.
>
> **Q4 Is it possible that the sweet spot can be too short for some long-range problems?**
>
> The "sweet spots" of depth in our experiments are computed using Lemma 2 and Theorem 2 for each specific CSBM configuration accordingly. For certain configurations, they could be as long as up to 14 layers for the node classification task (Figure 4A). However, the sweet spots are only optimal in the sense of noise-reduction and in principle may not be ideal for the entire learning problem when long-range information is needed. The theoretically derived "sweet spots" appearing short is precisely why oversmoothing can be a concern in practice.
>
> **Q5 Notation instruction for $\omega/\Omega$?**
>
> We adopted the standard notation for $\omega/\Omega$: we say $f(n)$ is $\omega (g(n))/\Omega (g(n))$ if there exists a constant $c > 0$ and an integer $n_0 \geq 1$ such that $f(n) > cg(n)/f(n) \geq cg(n)$ for every $n \geq n_0$.
>
> **Q6 Explain more about Figure 2?**
>
> Thanks for the question.
> - In 2A, we compare the optimal depth in practice (blue curve) against the theoretically predicted optimal range (gray bar) given by Lemma 2 and Theorem 2. Thus 2A shows that
>     - our theory can accurately predict the optimal depth for CSBMs;
>     - when the community structure is stronger (i.e. the $a$ on the x-axis is larger), the optimal depth is larger and oversmoothing happens later.
> - 2B&2C show that when the community structure is stronger, given the same set of node features, GNN can achieve better optimal performance (2C), and the poorer performance is not due to overfitting (2B).
>
> We have added more explanation in the updated manuscript to make the description of Figure 2 more clear.
>
> **Q7 How does PPNP perform at a shallow depth after all?**
>
> Thanks for the nice catch! Indeed, PPNP corresponds to an "infinite-depth'' APPNP and thus does not have a notion of depth, and our original wording was slightly confusing.  We have updated the manuscript for better clarity by deleting PPNP in the second sentence.
>
> We appreciate your questions and comments. Please let us know if you have any further questions.

---

> > ### Author Response · Authors · 2022-11-18
> > **References**
> >
> > [1]Kipf and Welling. Semi-supervised classification with graph convolutional networks. ICLR 2017.
> > [2]Li et al. Deeper insights into graph convolutional networks for semi-supervised learning. AAAI 2018.
> > [3]Oono and Suzuki. Graph neural networks exponentially lose expressive power for node classification. ICLR 2020.
> > [4]Chen et al. Simple and deep graph convolutional networks. ICML, 2020.
> > [5]Di Giovanni et al. Graph neural networks as gradient flows, 2022.

---

### Official Review · Reviewer_2Zpy · 2022-10-27

**Confidence:** 4
**Correctness:** 3
**Technical Novelty And Significance:** 2
**Empirical Novelty And Significance:** 2
**Recommendation:** 3

**Clarity, Quality, Novelty And Reproducibility:**

The current work is novel albeit in a highly specific set-up. The problem statement, motivation and proposed approach are clear.

**Strength And Weaknesses:**

Here below are some strengths of the current work :

1) The authors focus on an important problem in the GNN research community and make an effort to decipher it in this current work. Specifically the authors try to explain why over smoothing happen at a relatively shallow depth which is very important to understand.

2) The authors demonstrate the presence of counteracting effects of applying graph convolutions i.e., a mixing effect which homogenizes node representations and a de-noising effect which homogenizes node representations belonging to the same class. The authors also provide an estimate as to when one effect dominates the other.

Here below are some weakness of the current work :

1) The current work focusses on a simplified setup of two classes and uses Contextual Stochastic Block Model as the generative model which does not provide an accurate representation of real-world graphs.

2) The authors do not demonstrate how to measure over-smoothing as well as the two effects in practice. Without being able to 1) adequately measure and quantify the two effects and 2) demonstrate the authors can adequately explain over-smoothing in a general setting, the presented approach seems hypothetical.

3) The authors do not demonstrate the practicality of the current approach to the research community. Theoretical analysis is great however without being able to provide utility, the scope of the work is significantly reduced.

4) As the authors themselves point out, the current work's analysis is based on oracle classifier, however in practice, we work in a semi-supervised setting. Thus the utility associated with the current work is pretty limited.

**Summary Of The Paper:**

In this work, the authors focus on the over smoothing issue prevalent in GNN and demonstrate the presence of two underlying mechanisms which dictate when and how over smoothing issue crops up i.e., an undesirable mixing effect and a desirable de-noising effect. The authors demonstrate the efficacy of their approach via empirical results.

**Summary Of The Review:**

The authors focus on an important problem for the GNN research community and try to decipher it via breaking it down to two counteracting effects of applying graph convolutions. However the analysis is very focussed on a specific set-up and is not generic in nature. Additionally it makes strong assumptions and due to this and prior issues, the usefulness of the current work is pretty limited.

---

> ### Author Response · Authors · 2022-11-18
> **Response to Reviewer 2Zpy**
>
> Thank you for taking the time to review our paper. We are encouraged that you find our perspective and theory novel. Before providing detailed responses, we would like to first clarify the purpose of our theory-focused work and emphasize the significance of our theoretical contributions:
>
> Our main goal is to theoretically understand how oversmoothing happens in GNNs with a finite depth. To that end, we develop a new theoretical framework for studying the behavior of finite-depth GNNs based on the Contextual Stochastic Block Model (CSBM) and establish the first quantitative theory for oversmoothing in GNNs with a small, finite number of layers. While all models are a simplification of reality, our model quantitatively characterize the main mechanism behind oversmoothing, which we have further tested on real data following your suggestion (see below, 2).
>
> We address the detailed concerns below:
>
> **1. The two-class CSBM doesn't provide an accurate representation of real-world graphs.**
>
> - Thanks for the comment. Block models are developed as the theoretical abstraction to capture the community structure of real-world graphs while being amenable to precise theoretical analyses, thus becoming canonical testbeds to study clustering and community detection methods [1,2], including GNNs [3,4]. Moreover, as the community structure and graph diameter can be controlled by the hyperparameters of the CSBM, it allows us to demonstrate how these two features affect the occurrence of oversmoothing and thus to quantitatively characterize the main mechanism behind the phenomenon.
> - With similar techniques, our results can be extended to multi-class setups, which we leave for future work.
>
> **2. The authors do not demonstrate how to measure oversmoothing and the two effects in practice.**
>
> Thanks for the comment. Actually, how to define and measure oversmoothing quantitatively is a main question that we are addressing with our study. We explicitly measure the two effects via the inter-class difference between the intra-class means ($|\mu_2^{(n)} - \mu_1^{(n)}|$) and the intra-class variance ($(\sigma^{(n)})^2$). On CSBM graphs, we theoretically derive estimates of these two measures for any model depth (Theorems 1&2), which allow us to characterize oversmoothing in finite-depth GNNs. These theoretical results are validated by our numerical experiments to accurately predict the phenomenon in practice, as shown in Figure 2&4A.
>
> On real-world graphs, the two effects can also be measured empirically via the two metrics above and we observe a similar tradeoff between them. In the table below, we present the average pairwise L2-distance between the intra-class means and the average intra-class variance with respect to increasing model depth on Cora, CiteSeer and PubMed. We see that both quantities decrease as model depth increases, which is consistent with our theoretical analysis.  More detailed results with visualization are provided in the Appendix L.4 of the updated manuscript.
> |depth|Cora|CiteSeer|PubMed|
> |:-:|:-:|:-:|:-:|
> ||*mean var*|*mean var*|*mean var*|
> |0|1.064 16.848|1.086 29.597|.101 .153|
> |1|1.053 7.607|1.046 19.893|.090 .103|
> |2|.960 3.605|1.012 13.289|.085 .032|
> |3|.940 3.040|.997 13.354|.084 .043|
> |4|.890 2.215|.987 11.236|.080 .020|
>
> **3. The work provides no practical utility hence the scope is significantly reduced.**
> - The primary motivation of our work is indeed a theoretical one, but we respectfully disagree that this alone should render it not meaningful. Through our new analysis framework, we develop the first quantitative theory of oversmoothing in GNNs with finite depth, which offers new insights by identifying the tradeoff between the two competing effects of graph convolutions as the underlying mechanism.
> - Moreover, our work is practically relevant in at least two ways.
>   - First, our theory predicts the "sweet spots" for the choice of depth on the CSBM, which opens up the possibility of choosing it in a principled fashion instead of hand tuning.
>   - Second, in contrast with some conventional wisdom, our theory shows why the PPNP/APPNP models are unable to fundamentally solve the problem of oversmoothing, thus paving the way for a better usage of such models.
>
> **4. The analysis is based on an oracle classifier but in practice we have a semi-supervised setting.**
>
> - The use of the "oracle'' Bayes optimal classifier allows one to probe into the fundamental statistical limitation of models like GNNs. It is a well-established approach in the machine learning literature [5], including theoretical analyses of GNNs [3,4].
>
> - Moreover, our experiments show that the theoretical predictions based on the Bayes optimal classifier are very close to the empirical performance of a classifier trained in a standard semi-supervised setting in our case (Figure 2A&4A).
>
> We hope that our response can resolve your concerns and you may consider adjusting the assessment. Please let us know for any further questions.

---

> > ### Author Response · Authors · 2022-11-18
> > **References**
> >
> >
> > [1]Abbe. Community detection and stochastic block models: recent developments. Journal of Machine Learning Research, 2017.
> >
> > [2]Deshpande et al. Contextual stochastic block models. NeurIPS, 2018.
> >
> > [3]Baranwal et al. Graph convolution for semi-supervised classification: Improved linear separability and out-of-distribution generalization. ICML, 2021.
> >
> > [4]Wei et al. Understanding non-linearity in graph neural networks from the bayesian-inference perspective. NeurIPS, 2022.
> >
> > [5]Devroye et al. A probabilistic theory of pattern recognition, 1996.

---

### Author Response · Authors · 2022-12-11
**Concerns about the reviews**

While we appreciate a number of questions and suggestions made by the reviewers, we feel that the negative review given by reviewer 2Zpy is unjustified and unrelated to our theoretical results and contributions, and mostly due to a complete misunderstanding of the role of theory and theoretical analysis. Our paper is a theoretical analysis on oversmoothing in finite-depth graph neural networks (GNNs), where we provide the first quantitative, low-depth theory of oversmoothing in GNNs. Our goal is to develop better insights than the previous asymptotic analysis (which only shows that oversmoothing is inevitable for infinite-depth GNNs) into the problem---that is why we submitted the paper to the “Theory” track.

While reviewer 2Zpy acknowledges our theoretical novelty, the reviewer provides no comment on the theorem statements and proofs themselves. The weaknesses raised by the reviewer are centered around a misunderstanding of the role of theory and they question the practicality of our work. Obviously theoretical abstractions are not meant to replicate the full complexity of the real problem, but that does not mean theoretical analysis is not useful.

Our theory captures the essence of oversmoothing using simplified abstractions of the problem that focus only on relevant characteristics. As reviewer K5CA commented, while we make some standard simplifying assumptions, “enough structure is imposed on the synthetic graphs for the results to be meaningful.”


The authors

---

### Decision · Program_Chairs · 2023-01-20

**Decision:**

Accept: poster

**Justification For Why Not Higher Score:**

This paper has some limitations: (1) the model is restrictive, that is, only CSBM is analyzed, (2) the Bayes oracle classifier is analyzed instead of trained classifier. In addition to that, the statement to the existing work could be improved. For these reasons, this paper is not as strong as spotlight presentation.

**Justification For Why Not Lower Score:**

The paper gives a sharper analysis of the oversmoothing effect that has not been captured in the existing work. Such an analysis is beneficial to the community.

**Metareview: Summary, Strengths And Weaknesses:**

This paper gives more quantitative evaluation of the oversmoothing effect of GNNs so that it explains a fact that the oversmoothing phenomenon occurs in lower layers. For that purpose, the authors investigated Contextual Stochastic Block Model (CSBM) and gave quantitative convergence with respect to the number of layers. The theoretical finding is justified by some numerical experiments.

Strength: This paper gives a detailed analysis that characterizes the oversmoothing effect of GNNs in the setting of CSBM. It gives an explicit evaluation of the convergence rate. In particular, this paper gives a precise analysis of mixing effect and denoising effect which is helpful to understand the oversmoothing effect compared with existing Laplacian eigenvalue analyses. In particular, it shows that there appears a sweet spot to avoid the oversmoothing. This is a novel result.

Weakness:
- The paper only analyzes CSBM which is restrictive. It is not entirely clear whether the analysis in this paper gives a general insight to the practical settings. Indeed, there is a contradictive result between theory and experiments as indicated by the reviewer K5CA. It would be better if the analyses could be generalized to more general settings.
- The Bayes oracle classifier is analyzed as a simplification and the analysis can not be applied to training of GNNs.
- I (AC) personally think it would be misleading to state that the existing work fail to give a quantitative convergence rate because they show the oversmoothing effect only in the infinite layer asymptotics. For example, Oono&Suzuki (2020) shows the exponential convergence with respect to the number of layers and characterized the rate of convergence by the eigenvalues of the graph Laplacian. Moreover, they derived the rate of convergence for the Erdos-Renyi graph by evaluating the spectral gap (which is also utilized in the proof of Theorem 2 in this paper). Hence, the statement about existing work had better to be properly revised.

In summary, although this paper has some weakness, its theoretical contribution is indeed novel. It gives much more detailed characterization of oversmoothing effect than existing work while the setting is a bit restrictive. Hence, I recommend acceptance of this paper.

**Note From Pc:**

if the above contains the word "oral" or "spotlight" please see: "oral" presentation means -> notable-top-5% and "spotlight" means -> notable-top-25%. As stated in our emails, we are disassociating presentation type from AC recommendations

---

> ### Author Response · Authors · 2023-03-06
> **Response to AC**
>
> Dear Area Chair,
>
> We thank you for your efforts in summarizing the reviews and providing feedback, and truly appreciate that you find our work novel and meaningful to the community.
>
> However, we would like to address the third weakness point mentioned in your comment. Indeed, as you stated, prior studies such as the seminal work by Oono and Suzuki proved that graph signals converge exponentially to the dominant eigenspace of the graph Laplacian, and we had no intention at all to claim that it lacks a quantitative convergence analysis. Rather, we argued that their work was unable to fully explain the occurrence of oversmoothing at *finite* depths as it did not separate the (desirable) denoising effect from the (undesirable) mixing effect of graph convolutions, whose interplay underlies the onset of oversmoothing and is the basis of our quantitative analysis of the role of depth in finite-depth linear GNNs.
>
> Please let us know if you have any further questions, and thank you again for your time and consideration!
>
> The authors